# Highly conductive single-molecule junctions through electrocatalytic formation of benzyl-type Au–C bonds

Yaxuan Zhang[1,2,4], Kai Qu[1,2,4], Ting Pan[1,2,4], Yaqi Zhang[1,2], Leng Wang[1,2] & Hongliang Chen [1,2,3] ✉

Creating reliable molecular-scale electronic devices demands strong, stable connections between metal electrodes and organic molecules. A significant challenge is forming robust chemical bonds directly to gold electrodes, as gold is notoriously unreactive. Conventional methods for creating gold-carbon (Au–C) bonds are therefore limited. Here we demonstrate an electrocatalytic solution: using an applied voltage, we inject a single electron from a gold electrode into specific organic salts (pyridinium ions). This electron transfer breaks the salt apart, generating highly reactive carbon-based radicals. These radicals spontaneously form strong, direct covalent bonds (Au–C) with the gold surface. Using precise single-molecule measurements, we show this radical-mediated bonding creates exceptionally stable molecular junctions. Furthermore, these junctions exhibit excellent electrical conductivity across the molecule's core structure. This high conductivity arises because the direct Au–C bond allows efficient overlap of electron orbitals between the gold and the molecule. Our strategy provides a versatile and controlled way to build atomically precise, highly conductive interfaces between metals and organic components, advancing the design of functional molecular electronics through tailored covalent connections.

Modern microelectronics relies heavily on the properties of interfaces between two distinct components[1,2], to determine the efficiency of electron transport[3], as well as the overall reproducibility and device stability. For example, over the past few years, there has been increasing focus on the formation[4–14] of robust Au–C covalent bonds across the interface of single-molecule devices. The chemical, mechanical, and thermal stability of Au–C covalent bonds, along with their electronic properties, make them offer advantageous for achieving stronger coupling strength compared to other commonly used anchor groups such as amines[15,16], thiols[17], thiomethyls[18], and pyridines[19,20].

On account of the chemical inertness of Au electrodes, in-situ generation[21] of Au–C bonds relies on highly reactive reagents, enabling the targeted transfer of the carbon species onto the Au surface. A key aspect of this approach involves the careful installation, utilization, and eventual removal of protecting groups in π-conjugated conducting molecules, to achieve the desired complexity with controllable reactivity. To date, many methods (Fig. 1 and Table 1) have been developed to create an Au–C and Si–C bonded interface[4–14,22,23]. Firstly, terminal alkynes can form Au–C bonds either spontaneously[8,24,25], or through tetrabutylammonium fluoride (TBAF)-mediated deprotection of trimethylsilyl (TMS)-capped oligo(phenylene ethynylene)[9,10], with liberated $sp$-hybridized carbons creating robust interfacial Au–C contacts. Secondly, structurally defined $sp^2$-hybridized carbons enable direct Au–C bonding via nanogap-confined electrocatalysis, with representative methodologies including: (i) electroreduction of

[1]Stoddart Institute of Molecular Science, Department of Chemistry, Zhejiang University, Hangzhou, P. R. China. [2]ZJU-Hangzhou Global Scientific and Technological Innovation Center, Zhejiang University, Hangzhou, P. R. China. [3]Beijing National Laboratory for Molecular Sciences, Beijing, P. R. China. [4]These authors contributed equally: Yaxuan Zhang, Kai Qu, Ting Pan. ✉e-mail: hongliang.chen@zju.edu.cn

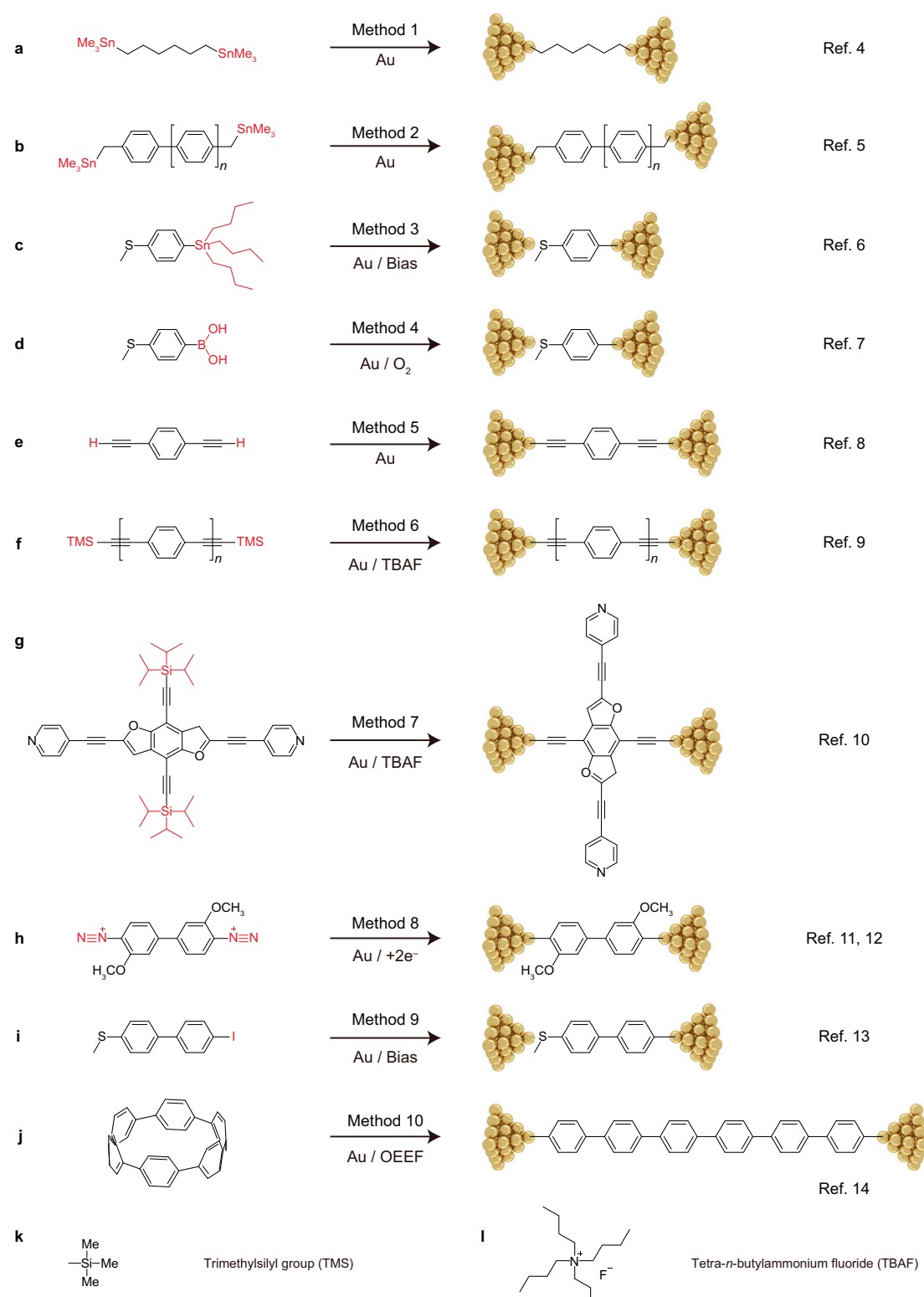

In-situ formation of covanlent Au–C bonding at the imterface

diazonium-terminated biphenylenes[11,12], (ii) oxidative addition of aryl iodides to Au[13], (iii) electric-field-catalyzed Au–C formation from organotin[6] or organoboron compounds[7], and (iv) electric-field-catalyzed ring-opening of cycloparaphenylenes[14].

While several other methods[18,26] have been reported for creating Au–molecule covalent bonding at the interfaces, achieving highly conductive Au–C interfaces remains fundamentally constrained by inherent orbital mismatching that directly limit electron delocalization efficiency. Venkataraman's seminal studies[4,5] established that only the $sp^3$-type Au–CH$_2$–Ph linkage enables effective $\pi$-system coupling at Au interfaces, enhancing conductance via frontier orbital hybridization. This mechanism aligns with prevalent synthetic strategies employing in-situ removal of tin-containing moieties, as demonstrated in prior interfacial engineering protocols (Fig. 2a). However, a potential drawback of this

**Fig. 1 | Summary of methods developed to create Au–C covalent bonding at the interface.** **a** (Method 1) Bis-trimethylstannane-containing alkanes react with gold to form covalent Au–C bonds in-situ at the interface by means of tin-gold reaction[4]. **b** (Method 2) In-situ Au–C covalent bonding at the interface by reaction of trimethyltin-capped oligophenylene with gold[5]. **c** (Method 3) In-situ *n*-butyl-substituted organotin cleavage to form a covalent bonding between gold and *sp*[2]-type carbon species[6]. **d** (Method 4) In-situ formation of Au–C covalent bonds occurs at the interface through electric potential-promoted oxidative coupling between gold and organoboronic acids[7]. **e** (Method 5) In-situ formation of Au–C covalent bonding between gold and *sp*-type terminal alkynes[8]. **f** (Method 6) Trimethylsilyl (TMS)-capped oligo(phenylene ethynylene)s reacts with gold after TBAF treatment to form covalent *sp*-type C–Au bonds at the interface[9]. **g** (Method 7) The triisopropylsilyl group undergoes in-situ cleavage with TBAF, facilitating complete transformation of N–Au linkages into C–Au bonds[10]. **h** (Method 8) In-situ formation of Au–C covalent bonds is achieved through electrochemical reduction of diazonium terminal groups on gold electrodes[11,12]. **i** (Method 9) In-situ formation of Au–C contacts via bias-promoted oxidative addition of aryl iodides to gold[13]. **j** (Method 10) In-situ generation of Au–C covalent bonds guided by an oriented external electric field (OEEF), facilitated by electric field-induced bond scission in cycloparaphenylenes[14]. **k** Chemical structure of the trimethylsilyl group (TMS). **l** Chemical structure of the tetra-*n*-butylammonium fluoride group (TBAF). Groups labeled in red in the figure represent leaving groups.

## Table 1 | Comparison of existing methods to establish Au–C covalent contacts [4–14,22,23]

| Method | Advantages | Disadvantages | Ref. |
|---|---|---|---|
| 1 | In-situ and stable | Highly toxic substrate (LD$_{50}$ < 50 mg kg$^{-1}$) | 4 |
| 2 | In-situ and stable | Highly toxic substrate (LD$_{50}$ < 50 mg kg$^{-1}$) | 5 |
| 3 | In-situ | Highly toxic substrate (LD$_{50}$ < 50 mg kg$^{-1}$) Prone to coupling reaction | 6 |
| 4 | Environmentally friendly | Prone to coupling reaction | 7 |
| 5 | Spontaneously | Unstable substrate | 8 |
| 6 | Stable and efficient | Ex-situ | 9 |
| 7 | Stable and efficient | Ex-situ | 10 |
| 8 | Environmentally friendly | Unstable substrate | 11,12 |
| 9 | High efficiency | Unstable substrate Prone to side reactions | 13 |
| 10 | Environmentally friendly | Limited substrate | 14 |
| 11 | Stable (Si–C bond) | Limited substrate (silicon) | 22,23 |

Stable: molecules can form covalent bonds at both ends with Au electrodes. Median lethal dose (LD$_{50}$): an indicator commonly used in toxicology to measure the toxicity of a substance. Environmentally friendly: the process replaces toxic substrates and enables reactions via green methods like electrocatalysis. Unstable substrate: The reactant has a short half-life and is prone to decomposition under ambient conditions. Limited substrate: the method is restricted to compounds with specific structures for constructing Au–C bonds, making it difficult to extend to diverse molecular systems.

approach is the toxicity[27] of organotin precursors. Besides, trace residual organotin reagent species (e.g., Bu$_3$SnH or Bu$_6$Sn$_2$) readily adsorb onto Au electrodes, impeding interfacial contact with π-conjugated wires and reducing junction formation yield significantly.

Electrocatalytic single-electron transfer (SET) activation of pyridinium scaffolds enables controlled radical generation through C–N bond homolysis[28–30], establishing a synthetic platform for controlled carbon radical liberation (Fig. 2b and Supplementary Fig. 1). The Katritzky salt—a redox-active pyridinium derivative[30,31]—mediates SET-driven fragmentation, producing synthetically useful radicals while regenerating pyridine byproducts. This mechanistic paradigm aligns with experimental advances in photoredox[32,33] and transition-metal catalysis[34,35], where SET strategies unlock transformations inaccessible under mild conditions.

In this research, we employed this electrocatalytic methodology to establish covalent Au–C connections in single-molecule devices (Fig. 2c). Specifically, this methodology involves using scanning tunneling microscope-based break junction (STM-BJ), where electron[36–39] from the STM tip served as the reductant in SET approach to (i) initiating the homolytic fragmentation of Katritzky salts, then (ii) enabling the release of carbon radicals, and eventually (iii) leading to the in-situ formation of benzyl-type Au–C covalent bond in single-molecule junctions. A series of Katritzky salts (Fig. 2d–f) were selected as model compounds to demonstrate this strategy using STM-BJ measurements in combination with ensemble electrochemical experiments in solution. This radical-mediated bonding creates a *sp*[3]-type Au–CH$_2$–Ph linkage in single-molecule junctions with near-resonant conductance (≈1 $G_0$) across 8 Å π-conjugated backbone. This investigation present herein introduces an electrocatalytic methodology for the formation of *sp*[3]-type Au–C covalent bonds in single-molecule devices with enhanced robustness and high conductance, establishing a foundation for the development of robust functional molecular devices in the near future.

## Results

### SET-induced homolysis of C–N bonds in Katritzky salts

To verify the feasibility of SET-induced homolysis at the single-molecule level, we conducted STM-BJ experiments on a series of *N*-functionalized Katritzky salts, namely PyBz–*n* (*n* = 1, 2, and 3), in MeCN solvent at a concentration of 0.1 mM. Upon SET from the tip to the compound (Fig. 3a), PyBz–*n* is expected to undergo fragmentation, resulting in the formation of two components, i.e., (i) a benzyl radical serving as the leaving group absorbed on Au surface, and (ii) a pyridine moiety, forming Au–S–Py$^n$–N–Au (*n* = 1, 2, and 3) junctions in-situ, featuring with a thiomethyl (–SMe) anchor at one end and a pyridine anchor at the other end. Therefore, measuring the conductance of the newly formed pyridine junctions can offer essential evidence for the presence of SET-induced homolytic fragmentation of C–N bonds in Katritzky salts.

We created one-dimensional (1D) conductance histograms from breaking traces of PyBz–3 collected at various bias voltages (Fig. 3b). In the bias voltage range of 0.1–0.3 V, the 1D conductance histograms displayed negligible conductance features (Fig. 3b, bottom), suggesting the lack of a stable molecular junction. A new conductance peak appeared gradually at around 10$^{-4.0}$ $G_0$ at a bias of 0.4 V, and its prominence increased as the bias voltage was raised up to 0.7 V (Fig. 3b, middle, and Supplementary Fig. 45). This phenomenon indicates the emergence of SET-induced formation of a new species, specifically Py–3, within the nanogaps. To validate this hypothesis, we synthesized Py–*n* (*n* = 1, 2, and 3) ex situ and performed STM-BJ experiments at a bias voltage of 0.7 V. Figure 3c compares the typical single-molecule breaking traces for PyBz–3 and Py–3 at a bias voltage of 0.7 V. These two compounds have exhibited clear and similar conductance signatures, both positioned around 10$^{-4.0}$ $G_0$.

Specifically, the two-dimensional (2D) conductance histogram of PyBz–3 exhibited a conductance plateau with a length of 0.8 nm at a bias voltage of 0.7 V (Fig. 3d), mirroring the conductance characteristic

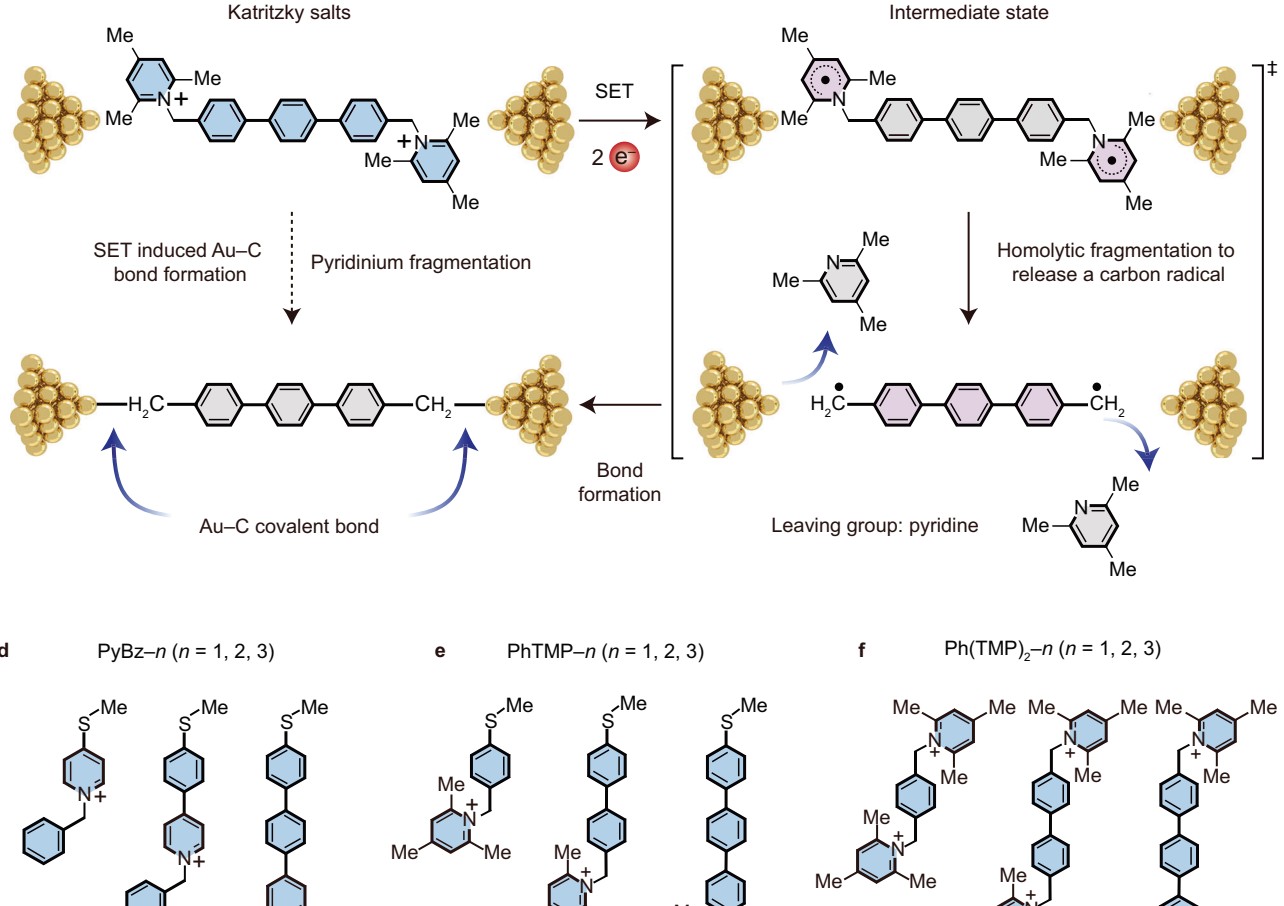

**Fig. 2 | Formation of Au–C covalent bonds. a** In-situ cleavage of C–SnMe$_3$ results in the generation of Au–C covalent bond. **b** SET-induced homolysis of Katritzky salts results in the generation of carbon radicals and pyridines. **c** Electrocatalytic formation of Au–C covalent bonds in single-molecule junctions. **d** Structural formulas of PyBz–n, used in this research (**n** = 1, 2, and 3). Counterions (Br⁻) are omitted for clarity. **e** Structural formulas of PhTMP–n, used in this research (**n** = 1, 2, and 3). Counterions (Br⁻) are omitted for clarity. **f** Structural formulas of Ph(TMP)$_2$–n, used in this research (**n** = 1, 2, and 3). Counterions (Br⁻) are omitted for clarity. The interior of the molecule is filled with blue for a positively charged molecule, purple for a free radical state, and grey for a neutral molecule.

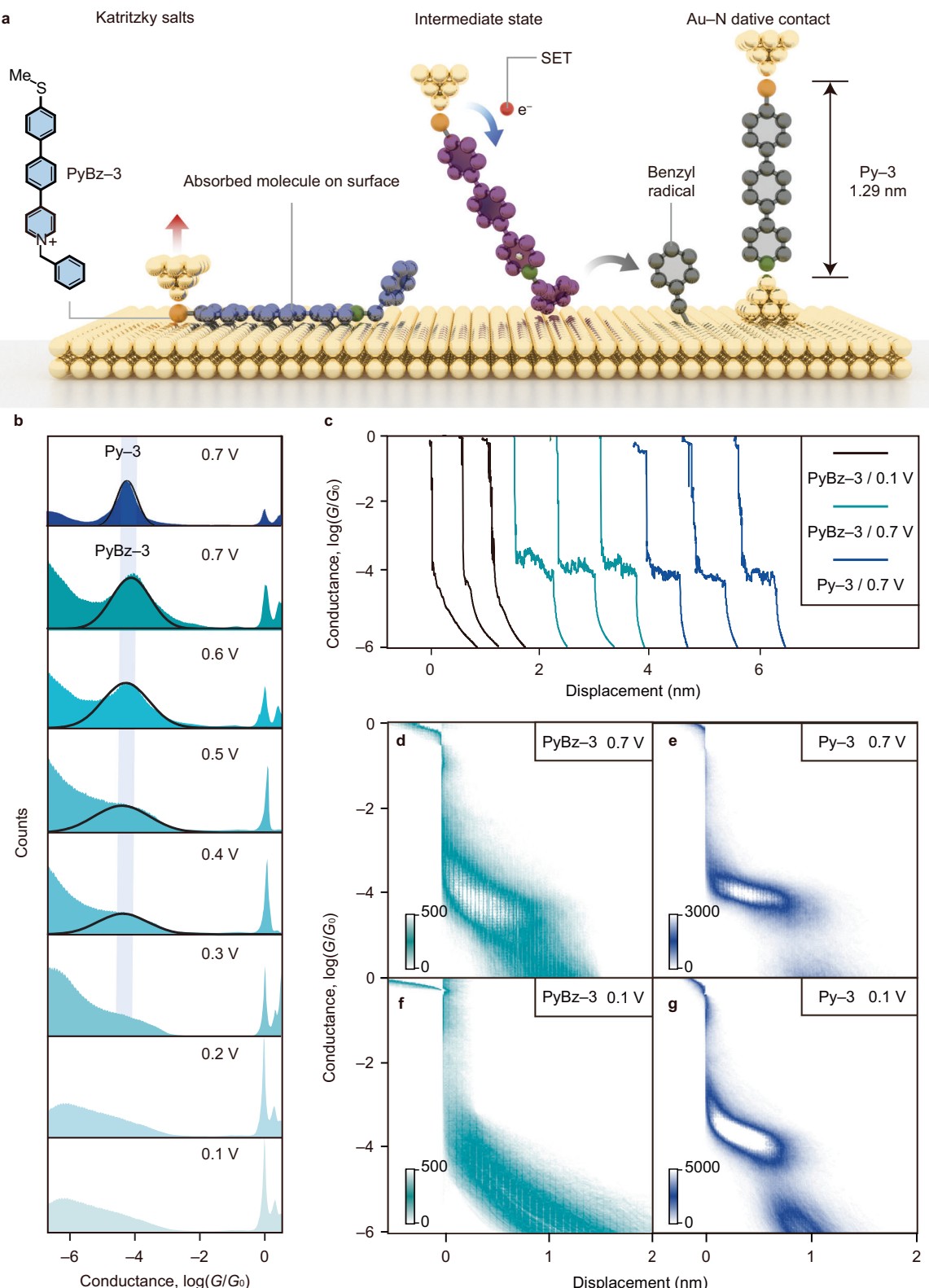

**Fig. 3 | In-situ formation of Py–3 through SET-induced homolysis of PyBz–3 in a single-molecule junction. a** Schematic illustration showing the SET-induced homolysis of PyBz–3 leading to the formation of Py–3 junction. **b** Top: The 1D conductance histograms of Py–3 and measured at a bias voltage of 0.7 V (blue). Bottom: 1D conductance histograms of PyBz–3 measured at increasing bias voltages from 0.1 to 0.7 V (cyan). **c** Typical individual conductance-displacement traces for PyBz–3 recorded at 0.1 (black) and 0.7 V (cyan), as well as Py–3 measured at 0.7 V (blue). **d** The 2D conductance histograms of PyBz–3 obtained in-situ and measured at $V_{Bias} = 0.7$ V. **e** The 2D conductance histograms of Py–3 prepared ex situ and measured at $V_{Bias} = 0.7$ V. **f** The 2D conductance histograms of PyBz–3 obtained in-situ and measured at $V_{Bias} = 0.7$ V. **g** The 2D conductance histograms of Py–3 prepared ex situ and measured at $V_{Bias} = 0.1$ V. The colour bar in the figure represents the number of counts per 1000 traces in the 2D matrix plot. Source data are provided as a Source Data file.

of Py–3 prepared ex situ (Fig. 3e). It is worth noting that, under the bias voltage of 0.1 V, the 2D conductance histogram of PyBz–3 (Fig. 3f) did not show a distinct conductance feature comparable to that of the 2D conductance histogram of Py–3 (Fig. 3g). All these results offer direct evidence of the SET-induced conversion from PyBz–3 to Py–3 at elevated bias voltages.

To assess the generality of the SET-induced conversion, we conducted bias-dependent STM-BJ experiments on PyBz–1 and PyBz–2 (Supplementary Figs. 43 and 44). These experiments resulted in comparable bias-dependent outcomes, wherein an increase in bias voltage led to the gradual emergence of a conductance signal consistent with the conductance signature of Py–$n$ ($n$ = 1, 2) prepared ex situ. We find that the conductance trends for both families are nearly linear against molecular length on the plot (Supplementary Fig. 46), with a decay constant ($\beta$) of 3.30 nm$^{-1}$ for PyBz–$n$ under a bias voltage of 0.7 V, which is similar to the $\beta$ value (3.14 nm$^{-1}$) of Py–$n$ at a bias voltage of 0.1 V. By combining the density functional theory (DFT) with the nonequilibrium Green's function (NEGF), the transmission spectra of the Py–$n$ molecular junctions were computed (Supplementary Fig. 58a). The theoretical $\beta$ value was determined to be 3.30 nm$^{-1}$ (Supplementary Fig. 58b), matching closely the experimental value. These findings provide supporting evidence supporting the occurrence of SET-induced homolytic fragmentation of Katritzky salts of PyBz–$n$, leading to the formation of neutral Py–$n$ wires in-situ.

## SET-induced formation of Au–C covalent bonds

To achieve the Au–C contact in single-molecule junctions, we synthesized PhTMP–$n$ ($n$ = 1, 2, and 3) as the target compounds, incorporating 2,4,6-trimethylpyridine (TMP) as the leaving group. In comparison to PyBz–$n$, PhTMP–$n$ exhibits the following structural characteristics. Upon SET from the tip to the compound (Fig. 4a), PhTMP–$n$ undergoes a fragmentation, resulting in the formation of two components, i.e., (i) a neutral TMP acting as the leaving group absorbed on Au surface, and (ii) a carbon radical intermediate, donated as Ph–$n$· ($n$ = 1, 2, and 3), featuring with a –SMe anchor at one end and a newly formed benzyl radical at the other end. Subsequently, the Ph–$n$· radical interacts with the Au electrode to establish Au–S-Ph$^n$-C–Au junctions in-situ within the nanogap.

Control compounds PhSn-$n$ ($n$ = 1, 2, and 3) were prepared, each featuring a tributyltin group (Bu$_3$Sn) at one terminus and a thiomethyl group (SMe) at the opposing end. PhSn-$n$ were known to form Au–C bond with an initial homolytic dissociation of a C–Sn bond on the Au surface, followed by known radical processes[4–6,40–44]. The in-situ generated molecular junctions exhibit structural identity to those constructed from PhTMP–$n$ through SET-mediated homolytic cleavage of C–N bonds. Figure 4b directly compares the 1D conductance histograms of PhTMP–$n$ measured at 0.6 V (top panel) with PhSn-$n$ datasets acquired at 0.1 V (bottom panel), demonstrating identical conductance peak positions (Supplementary Figs. 47, 48, and 49). The decay constant ($\beta$ = 4.8 nm$^{-1}$) determined for PhTMP–$n$ at 0.5 V closely matched the value ($\beta$ = 4.6 nm$^{-1}$) for PhSn-$n$ at 0.1 V (Supplementary Fig. 50).

A similar trend emerged in the 2D conductance distributions (Fig. 4c). Using PhTMP–3 and PhSn-3 as model compounds, the 2D conductance histogram demonstrated clearly resolved conductance plateaus, i.e., PhTMP–3 displayed a conductance plateau at ≈10$^{-4.0}$ $G_0$ with 0.9 nm length at 0.6 V (Fig. 4c, top panel), while PhSn-3 demonstrated identical conductance (≈ 10$^{-4.1}$ $G_0$) and plateau length (0.9 nm) at 0.1 V (Fig. 4c, bottom), after in-situ elimination of tributyltin groups. This robust correlation between the two molecular systems supports the reliable in-situ formation of Au–S-Ph$^n$-C–Au junctions with well-defined Au–C interfacial bonding configurations.

The asymmetric Au–S-Ph$^n$-C–Au junctions with unilateral Au–C bonds did not achieve high-conductance transport. Theoretical simulations on the Au–S-Ph$^1$-C–Au junction revealed no near-resonant

transmission near the Au Fermi level[5,45]. Instead, transmission peaks emerge at 0.3–0.5 eV (Fig. 4d), corresponding to the gateway state localized at the Au–C interface, as evidenced by isosurface analysis of the transmission eigenstate (Fig. 4d inset). This state exhibits strong hybridization with the molecular $\pi$-backbone and Au electrode orbitals. Projection eigenstate analysis also attributes the low conductance to weak interfacial coupling at the Au–S terminus. Moreover, identical transport behavior in Au–S-Ph$^2$-C–Au and Au–S-Ph$^3$-C–Au junctions (Supplementary Figs. 59 and 61) confirm the dominant role of interfacial bonding in governing the electron transport behavior.

## Capturing the intermediate radical species

To confirm the involvement of carbon radicals in the junction formation process induced by SET, we first of all conducted (Supplementary Figs. 54 and 55) cyclic voltammetry (CV) analysis on the PhTMP–$n$ series compounds. The experiments were performed in a 10 mM solution of KBF$_4$ in acetonitrile, with 0.5 mM of the target compound. The PhTMP–1 displays a single irreversible reduction peak at ≈–1.10 V, indicating rapid SET reactions of the target compound at the working electrode. Similarly, PhTMP–2 and PhTMP–3 exhibit (Supplementary Fig. 54) irreversible reduction peak at –1.21 and –1.38 V, respectively. Furthermore, UV–Vis spectroelectrochemical experiments were performed (Fig. 4e and Supplementary Fig. 56) using the solution obtained from the CV experiments at a specific reduction potential and reaction time. At low reduction potentials (≤ –0.6 V), PhTMP–1 exhibits (Fig. 4e) a dominant absorption at around 260 nm, with no peak detected above 280 nm. At a potential of –0.8 V, a new peak appears gradually at around 300 nm, reaching its maximum intensity at –1.0 V. As the constant potential of –1.0 V is applied over time, the absorption around 300 nm increases (Fig. 4f) gradually. This observation indicates that PhTMP–1 undergoes SET-induced homolysis, resulting in the generation of Ph–1· benzyl radicals and electron delocalization. It is important to note that in electrochemistry-related analyses, the voltage is applied using Ag/AgCl as the reference. In STM-BJ measurements, a bias voltage is applied across the source and drain electrodes, which differs from the voltage used in electrochemical measurements.

To verify that the newly formed intermediates are radical species, we performed EPR spectroscopy on PhTMP–$n$ ($n$ = 1, 2, and 3) following controlled potential electrolysis, using DMPO as a radical trapping agent. All three compounds have displayed (Fig. 4g red curves) clear EPR signals with hyperfine splitting patterns that are consistent with the simulated result (Fig. 4g black curve). The spectroelectrochemical and EPR spectroscopic studies demonstrate that Ph–$n$· radicals act as key intermediates in the SET-mediated homolysis of PhTMP–$n$, leading to Au–C covalent bond formation in STM-BJ experiments. This phenomenon establishes that SET-generated carbon radicals in molecular junctions undergo interfacial reactions with Au electrodes to form defined covalent contacts.

## Formation of Au–C covalent bonds at both termini

To address the suppressed conductance caused by weak Au–S interfacial coupling, we designed and synthesized Ph(TMP)$_2$–$n$ ($n$ = 1, 2, and 3) bearing TMP leaving groups at both termini of $\pi$-conjugated backbones, aiming to construct Au–C-Ph$^n$-C–Au molecular junctions with dual benzyl-type Au–C covalent bonds. However, the large steric hindrance of TMP groups on both ends of Ph(TMP)$_2$–$n$ compromises electrode-molecule coupling during the tip-substrate opening process in STM-BJ experiments, significantly reducing SET efficiency and preventing effective in-situ formation of Au–C-Ph$^n$-C–Au junctions. In contrast to the junction opening process, tip-mediated mechanical force applied during the junction closing process drives the contacts between the tips and TMP groups, thereby enhancing SET efficiency while facilitating in-situ Au–C bonds formation at the interface.

We achieved direct characterization of interfacial chemical reactions during closing processes[46]. Fig. 5b–d displays the 2D conductance-

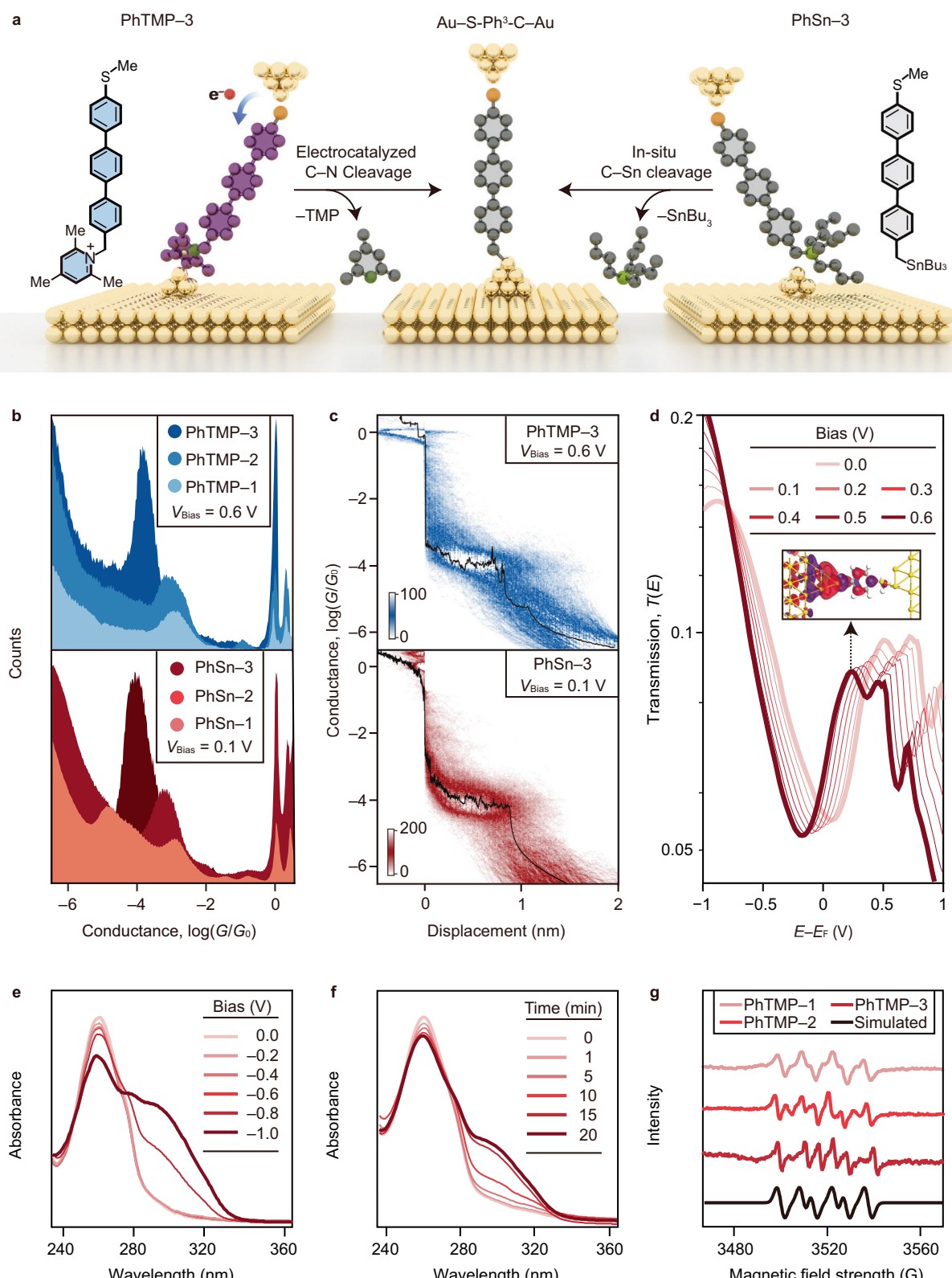

displacement histograms of the closing process for Au–C-Ph$^n$-C–Au junctions formed in-situ from Ph(TMP)$_2$–$n$ ($n = 1$, 2, and 3) precursors. During junction closure, a 0.1 V bias suffices to create Au–C-Ph$^1$-C–Au and Au–C-Ph$^2$-C–Au junctions, whereas Au–C-Ph$^3$-C–Au junction formation requires a higher voltage of 0.3 V. The relative displacement distributions for Au–C-Ph$^n$-C–Au series obtained from the closing process are demonstrated in Fig. 5b–d insets. Considering a jump-to-contact behavior of Au electrodes during the closing process with a

0.2 nm calibration distance applied[46], the molecular lengths derived from Au–C-Ph$^n$-C–Au ($n = 1$, 2, and 3) are 0.8, 1.2, and 1.6 nm, respectively. From the 1D conductance histogram (Fig. 5e), it is evident that Au–C-Ph$^n$-C–Au junctions with Au–C covalent bonds at both ends show high-conductance values reaching $10^0$, $10^{-1.0}$, and $10^{-1.7}$ $G_0$—with one, two, and three phenylene rings in the backbones, respectively.

To elucidate these experimental findings, we conducted DFT-based calculations[47–49] employing Au–C-Ph$^n$-C–Au molecular junctions

**Fig. 4 | Comparison of two in-situ methodologies of forming Au−C bonds in a single-molecule junctions. a** Schematic illustration showing the in-situ formation of Au–S-Ph³-C–Au junctions through electrocatalyzed cleavage C–N bond (left) or in-situ cleavage C–Sn bond. **b** The 1D conductance histograms of PhTMP–*n* recorded at a bias voltage of 0.6 V (top) and PhSn–*n* datasets recorded a bias voltage of 0.1 V (bottom). **c** The 2D conductance histograms of PhTMP–3 recorded at bias voltages of 0.6 V (top), and the 2D conductance histograms of PhSn–3 recorded at bias voltages of 0.1 V (bottom). The colour bar in the figure represents the number of counts per 1000 traces in the 2D matrix plot. **d** Calculated transmission spectra for PhTMP–1 under increasing bias voltages from 0 to 0.6 V. Inset

shows the position of transmission eigenstate corresponding to Au−C gateway state at 0.32 eV. **e** UV–vis spectroelectrochemical analysis of PhTMP–1 under progressively reducing potentials (0 to –1.0 V) revealed a characteristic absorption band at ≈300 nm, consistent with radical species (Ph–1˙) generation under cathodic conditions. **f** Absorption changes at ≈300 nm were monitored under a constant reduction potential of −1.0 V with incremental time extensions up to 20 min. **g** Electron paramagnetic resonance (EPR) spectra of Ph–*n*˙ radicals were obtained after controlled potential electrolysis using 5,5-dimethyl-1-pyrroline-*N*-oxide (DMPO) as a radical trapping agent. Source data are provided as a Source Data file.

with dual Au–C covalent anchoring interfaces (Supplementary Figs. 60 and 62). Transmission spectra analysis reveals a transition from near-resonant transport for Au–C-Ph¹-C–Au to non-resonant tunneling in longer molecules (Fig. 5f), consistent with prior reports[5,45]. Theoretical analysis yields a $\beta$ value of 3.88 nm⁻¹, demonstrating agreement with both our experimental value (4.65 nm⁻¹) and previously reported data (4.96 nm⁻¹) for analogous systems (Fig. 5g). The congruent exponential decay profiles across all three datasets provide robust theoretical validation of the experimental observations.

## Discussion

In this study, we utilized the technique of single-electron transfer-induced homolytic fragmentation of Katritzky salts to liberate benzyl-type carbon radicals and achieve the in-situ formation of Au–C contacts in single-molecule junctions. Through UV–Vis spectroelectrochemical experiments and electron paramagnetic resonance spectroscopy, we have identified the benzyl radical as a critical intermediate in the process of single-electron transfer. This radical-mediated bonding creates a $sp^3$-type Au–CH₂–Ph linkage in single-molecule junctions with near-resonant conductance across 8 Å $\pi$-conjugated backbones. These results highlight the efficacy of our electrocatalytic methodology for producing atomically defined metal-molecule interfaces toward the future development of molecular circuits with enhanced robustness and high conductance.

## Methods

### Materials

All reagents were used as received unless specified. Anhydrous acetonitrile (MeCN) for electrolysis and UV–Vis was purchased from Energy Chemical (anhydrous, 99.9%). Thin layer chromatography (TLC) employed silica gel 60 F254 (E. Merck), while column chromatography utilized a Biotage® Selekt system with RediSep Rf Gold® silica.

### Characterizations

Nuclear magnetic resonance (NMR) spectra were recorded on a Bruker Avance 600 spectrometer (¹H: 600 MHz, ¹³C: 151 MHz), with chemical shifts referenced to residual solvent peaks (DMSO-d6: $\delta_H = 2.54$ ppm, $\delta_C = 39.5$ ppm; CDCl₃: $\delta_H = 7.26$ ppm, $\delta_C = 77.2$ ppm). High-resolution mass spectra (HRMS) were obtained using a Bruker timstof pro mass spectrometer.

### General procedures for the synthesis of Katritzky salt salts

**General procedure for the synthesis of PyBz–*n*.** A mixture of 4-(methylthio)pyridine (5.0 mmol) and benzyl bromide (5.5 mmol) in acetone (20 mL) was stirred under N₂ at 90 °C for 12 h in a two-necked flask. After cooling to room temperature, the solvent was removed under reduced pressure. The crude product was filtered, washed with dichloromethane (3 × 10 mL), and dried under vacuum to afford PyBz–1 as a white solid in ≈96% yield.

**General procedure for the synthesis of PyTMP–*n*.** A solution of (4-(methylthio)phenyl)methanamine (1.0 mmol) in anhydrous isopropyl alcohol (30 mL) was treated with 2,4,6-trimethylpyrylium salt (1.1 mmol). The reaction mixture was stirred at 70 °C for 8 h under N₂,

concentrated under reduced pressure. The resulting residue was further purified through flash chromatography on a silica gel column using MeOH/CH₂Cl₂ ($v/v = 1$:10) as the eluent, yielding PyTMP–1 as a pale-yellow oil with a yield of ≈26%.

**General procedure for the synthesis of Py(TMP)₂–*n*.** To a solution of 1,4-bis(bromomethyl)benzene (2.0 mmol) in anhydrous MeCN (20 mL) was added 2,4,6-trimethylpyridine (20 mmol). The mixture was stirred at 90 °C for 12 h under N₂. After cooling to room temperature, the solvent was removed *in vacuo*. The crude solid was filtered, washed with Et₂O (3 × 15 mL), and dried under vacuum. This intermediate was dissolved in H₂O (30 mL), treated with NH₄PF₆ (4.4 mmol), and stirred vigorously for 30 min. The precipitate was collected by filtration, washed with H₂O (2 × 10 mL) and Et₂O (10 mL), then dried to afford Ph(TMP)₂–1 as a white solid with a yield of ≈82%.

### Electrochemical, UV–Vis spectroelectrochemical, and EPR spectroscopies

Cyclic voltammetry (CV) was performed at ambient temperature in N₂-purged MeCN using a CHI660E electrochemical workstation. Measurements used analyte solutions (0.5 mM in MeCN, 10 mM KBF₄ supporting electrolyte), a glassy carbon or Au working electrode, a Pt wire counter electrode, and an Ag/AgCl (3 M NaCl) reference electrode. For UV–Vis spectroelectrochemistry, solutions from CV experiments at specific reduction potentials and times were analyzed. Controlled potential electrolysis (CPE) was conducted in a modified UV–Vis cuvette using a Pt mesh working electrode, Pt wire counter electrode, and Ag/AgCl reference. UV–Vis spectra (Shimadzu UV-2600) were recorded in-situ after 1 min electrolysis.

Electron paramagnetic resonance (EPR) measurements at X-band (9.8 GHz) were performed using a Bruker A300, equipped with an ER4102ST resonator. EPR Samples were prepared by controlled potential electrolysis (CPE) experiments at a reduction potential of −1.0 V for 30 min, using DMPO (0.4 M in MeCN, 0.1 mL DMPO to 0.4 mL sample) as the spin-trapping reagent. The resulting adducts were transferred to 1 mm outer diameter capillary tubes for measurement under conditions of 1 G magnetic field modulation amplitude and 20.17 mW microwave power.

### STM-BJ measurements

**Sample preparations.** Single-molecule conductance was measured using a customized X-TECH STM-BJ system. Au substrates were prepared by evaporating chromium Cr/Au layers with controlled thickness of 5/100 nm on a silicon substrate with 300-nm-thick SiO₂ layer (Suzhou Research Material Microtechnology Co., Ltd.). Au tips were electrochemically etched from Au wire (99.998%, 0.25 mm diameter, Beijing Jiaming Platinum Nonferrous Metals Co., Ltd.). Target compound solutions—PyBz–*n*, PhTMP–*n*, Ph(TMP)₂–*n* (0.1 mM in H₂O), and PhSn–*n* (0.1 mM in TCB)—were dispensed (20 μL) onto Au substrates using a pipette before measurements. Conductance traces were collected at room temperature (20 °C) without selection.

**Conductance measurements.** During the STM-BJ measurement, the positioning and movement of the Au tip are controlled by a stepper

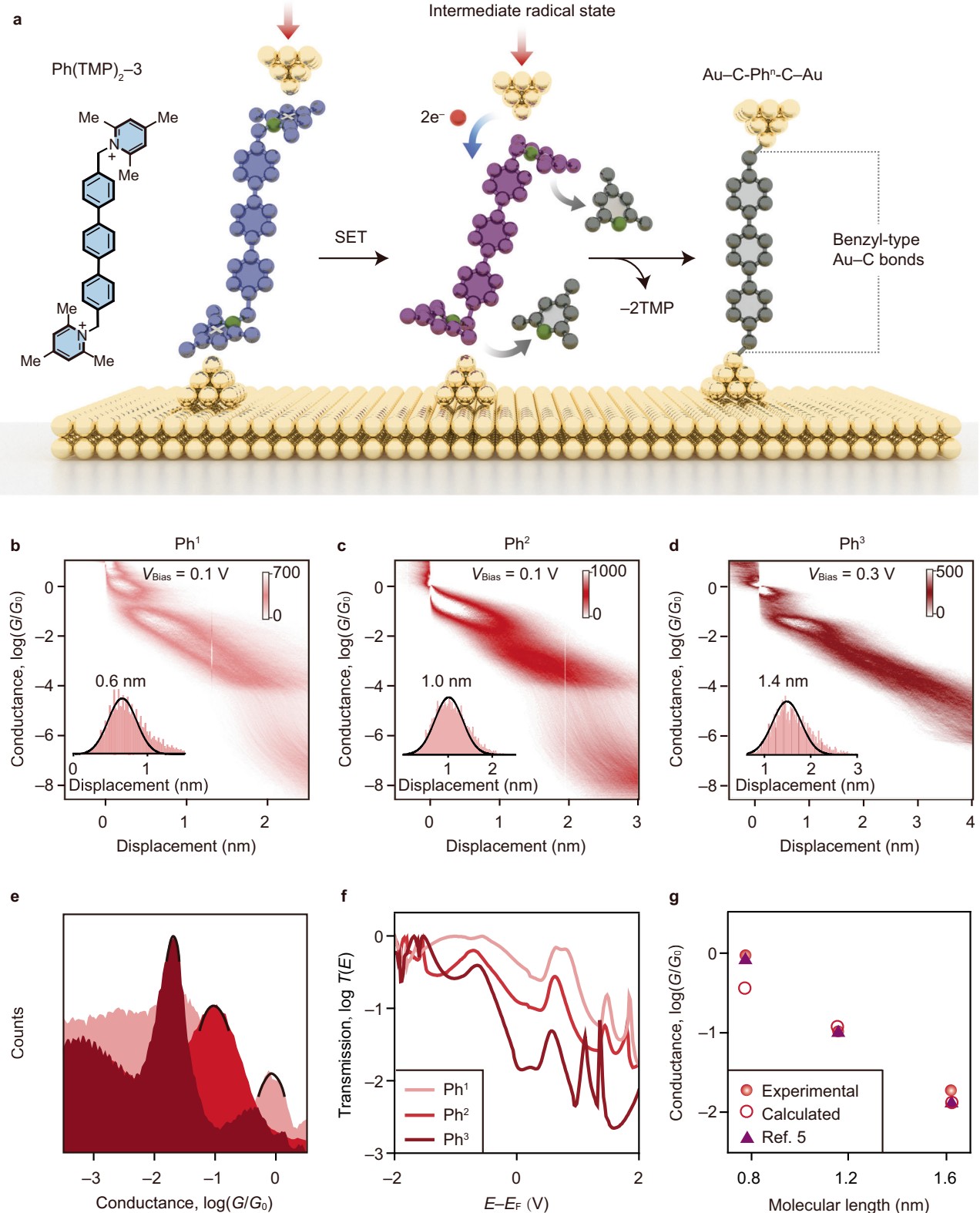

**Fig. 5 | Formation of Au–C covalent bonds at both termini in single-molecule junctions. a** Schematic illustration showing the electrocatalytic formation of benzyl-type Au–C bonds at both termini of during the closing processes. **b–d** The 2D conductance-displacement histograms constructed from the closing processes for the Au−C-Ph$^n$-C−Au junctions. Insets show the corresponding relative distance distributions. **e** 1D conductance histograms constructed from the closing processes showing the dominant conductance signals for the Au−C-Ph$^n$-C−Au junctions. **f** Calculated transmission spectra for Au−C-Ph$^n$-C−Au junctions. **g** Conductance versus molecular length determined from experimental measurements (red dots), theoretical calculations (red circle), and benchmark from Ref. 5. The colour bar in the figure represents the number of counts per 1000 traces in the 2D matrix plot. Source data are provided as a Source Data file.

motor (Harmonic Drive, LA-30B-10-F) and piezo stack (Thorlabs, PC4FL). The approximate position of the Au tip is controlled using a motor, allowing it to approach the substrate with an accuracy of less than 1 μm. Subsequently, the accurate movement of the tip is controlled by a piezo operating at a speed of 20 nm s$^{-1}$. With a maximum driving voltage of ±10 V, the piezo stack can control the displacement of the tip precisely with a variation range of ±15 %.

To ensure proper feedback control for conductance measurements, upper and lower limits of the current are defined, which is essential for maintaining the current within the desired range. As the tip moves upward, an Au–Au atomic contact is established between the tip and the substrate. Then the tip is further pulled, leading to the rupture of the contact. This action brings the target compounds in the nanogap into contact with the tip and the substrate, forming a single-molecule junction in which the conductance of the molecule is measured. Further pulling of the tip breaks the connection between the molecule and electrodes, causing the current to drop below the limit (1 pA) and become undetectable by the current amplifier. Throughout the entire process of contact rupture and reconnection, the bias is adjusted from 0.1 to 0.7 V. Real-time conductance is recorded using a customized I–V converter capable of sampling the conductance at a rate of 20 kHz.

**Data analysis.** In order to determine the most probable conductance and stretching distance during the measurements of single-molecule conductance, thousands of individual breaking traces were collected. A statistical approach without any data selection was employed for analysis. In order to construct one-dimensional (1D) and two-dimensional (2D) conductance-displacement histograms, all individual traces were collected with a bin size of 1100 for $\log(G/G_0)$ ranging from −10 to +1, and a bin size of 1000 for $\Delta z$ ranging from −0.5 to 3 nm. The peak shift was determined using Gaussian fitting, which represents the most probable molecular conductance. The 2D conductance-displacement histograms were generated by aligning with a relative zero point ($\Delta z = 0$) at 0.5 $G_0$. The relative stretching distance ($\Delta z$) was determined from the position where the conductance is 0.5 $G_0$ to the molecular conductance region just before the end of the molecular plateau. The peak in the histogram signifies the most probable length of the plateau. In order to calculate the absolute displacement ($z^*$), which correlates with the most probable length of the molecular junction, the relative displacements were adjusted by adding the snapback distance ($\Delta z_{corr}$) to the relative displacement ($\Delta z$), specifically, $z^* = \Delta z + \Delta z_{corr}$. Based on previous findings, $\Delta z_{corr}$ was experimentally determined to be 0.5 ± 0.1 nm.

**DFT calculations**

Firstly, the structural optimization of Py–$n$ was carried out using B3LYP-D3/6-311 G(d,p) basis set in the Gaussian 16 package. The structural optimization of Au-Ph–$n$ including Au–C covalent bonds were carried out using MO6L/Def2SVP basis set in the Gaussian 16 package.

Secondly, the optimized molecular structures were then bridged between two Au electrodes to construct the single-molecule devices. In order to build devices containing the Au–molecule–Au sandwich model, the surface of the Au electrodes was shaped into a pyramidal configuration. The top Au atom was coordinated by the terminal S atom of the molecule, while the bottom Au atom was coordinated by N atom (Supplementary Fig. 57) or covalently bonded with C atom (Supplementary Figs. 59 and 60).

Thirdly, we conducted geometric optimization for the devices by allowing relaxation in distances between all atoms within the bridge molecules as well as those between top and bottom Au electrodes. To expedite calculations, the scattering and extension area of the electrodes are maintained rigid, with the emphasis placed on optimizing the adsorption configuration between the electrodes and the

anchoring groups. The geometry optimization and transmission spectra calculation of the devices were implemented in Quantum ATK Q-2022.03 software. We adopted the generalized gradient approximation (GGA) Perdew-Burke-Ernzerhof (PBE) exchange-correlation functional, Fritz-Haber-Institute (FHI) pseudopotential combined with single zeta polarized group for Au atoms, and FHI pseudopotential combined with double zeta polarized group for other atoms.

## Data availability
The data that support the findings of this study are available from the corresponding author upon request. Source data are provided with this paper.

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

## Acknowledgements

The research at Zhejiang University was supported by the Fundamental Research Funds for the Central Universities (226-2024-00014). The authors also thank the National Natural Science Foundation of China (22273085, T2422020), Zhejiang Provincial Natural Science Foundation of China (LZ24B020004, LR25B020001), and Beijing National Laboratory for Molecular Sciences (BNLMS2023010) for the financial support. The authors would like to thank AI+ High Performance Computing Center of ZJU-ICI. We also thank Dr. Mowei Zhou from the Chemistry Instrumentation Center at Zhejiang University for their technical support.

## Author contributions

H.C. designed the experiments and supervised the project. K.Q., T.P., and Y.-Q.Z. were responsible for molecular synthesis and characterization. Y.-X.Z. carried out the single-molecule experiments and analyzed the data. Y.-X.Z. and L.W. performed the device fabrication, characterizations, and theoretical calculations. H.C., Y.-X.Z., K.Q., and T.P. wrote the manuscript with inputs from all authors. All the authors discussed the results and commented on the manuscript.

## Competing interests

The authors declare no competing interests.
