## [Transparent Peer Review file · Nature Communications]

Highly conductive single-molecule junctions through electrocatalytic formation of benzyl-type Au–C bonds

Corresponding Author: Professor Hongliang Chen

Version 0:

Reviewer comments:

Reviewer #1

(Remarks to the Author)

Since their discovery in the 1970s, Katritzky salts have become one of the most significant classes of building blocks in organic synthesis. These bulky pyridinium salts can participate in virtually any known reaction, alkylation, arylation, alkenylation, alkynylation, carbonylation, sulfonylation, and borylation. The current report expands their use to molecular electronics, specifically in the context of forming Au–C bonds. It should definitely feature in a good quality journal like *Nature Communications*.

One oversight by the authors of this technically very sound paper is that there is no mention of the very first paper describing how to make Au–C based molecular junctions. It was in a not-so-visible journal, but nonetheless, since it was the first, it should be cited and discussed (Pla-Vilanova, P.; Aragonès, A. C.; Ciampi, S.; Sanz, F.; Darwish, N.; Diez-Perez, I., The spontaneous formation of single-molecule junctions via terminal alkynes. *Nanotechnology* 2015, 26, 381001).

The paper is well researched and presented, and my only remark is on the possible involvement of the solvent. Is it possible that the reaction is indeed spontaneous (or become even more so) if a system that favors electron transfer and screens Coulombic interactions between surface holes and electrons is selected. I am aware this is not a semiconducting substrate, but e/h pairs form transiently even in metals, and it is possible that the time scale of the surface reaction is comparable to the lifetime of e/h pairs in Au. A discussion on how to test for this “hidden” spontaneous (electrochemical in nature) surface reactions was described by J. Gooding about 10 years ago in *Phys. Chem. Chem. Phys.* 2014, 16, 8003.

A pleasure to read, not very common to see images and data of this high quality.

Reviewer #2

(Remarks to the Author)

The manuscript “Electrocatalytic Au–C bond formation in single-molecule devices” by Yaxuan Zhang et al. describes the charge transport through single molecules using a scanning tunneling microscope-based break junction (STM-BJ) technique. The authors report that the injection of electrons by STM-BJ into molecular systems consisting of N-functionalized Katritzky salts, lead simultaneously, to the scission of a C–N chemical bond releasing pyridine molecules and benzyl radicals that form covalent Au–C bonds with Au electrodes. The strong molecule-to-electrode anchoring and the understanding of the charge transport at the individual molecular level has significant appeal for the progress of single-molecule-based electronic devices. Nonetheless, there are several points that the authors should address to support their statements. The comments are detailed below.

1. The authors mention in the introduction and Supplementary Information (SI) that up to date only four methods have been developed to form covalent Au–C bonds. Regarding this aspect, the literature review is outdated. The methodologies for forming C–metal covalent bonds have moved on recently. The description of this field should be refreshed.

2. The authors claim that the dissociation of the C–N bonds of the presented molecules occurs during the conductance measurements by STM-BJ. All the conductance curves presented in the manuscript and the SI show no clear evidence of such reactions. STM techniques are very sensitive to any molecular changes in the junction, even as geometrical conformations. The conductance traces and the peak visible in the conductance histograms prominent at certain bias voltages, have a standard appearance for the conductance through molecular junctions. The negligible conductance features at bias voltages close to the Fermi level may be related to the lack of molecular states at the corresponding

energies.

3. The similarities of the conductance features of PyBz-n (that are supposed to become Py-n) and the ex-situ prepared Py-n molecules should appear at similar bias voltages. The observed difference, that may be related to a different electronic structure, supports the previous comment. The PyBz-n and PhTMP molecules may have junction conformations similar to the ones before the reaction sketched in the left panels of the Supplementary Figures 14(a), 15(a), 16(a), 18(a), 19(a), 20(a). Thus, the charge transport may occur mostly through Py-n and Ph-n moieties where the orbitals that dominate the transport may be located, which may not have any weight on the Bz and TMP groups. Examples of similar scenarios can be found in the literature.

4. This manuscript focuses on the formation of covalent Au-C bonds. It is not clear why the authors present the Au/PhTMP-n/GaOx/EGaIn monolayer junctions in the last part of the manuscript, that is not directly related to the main focus of the manuscript and the description is overreached with links to SI. It gives the impression that the corresponding part is introduced to expand the length of the manuscript.

Reviewer #3

(Remarks to the Author)

The authors present an approach for forming covalent Au-C bonds in single-molecule devices through single electron transfer (SET) from gold electrodes to pyridinium salts. This methodology introduces some elements of novelty, particularly in the context of electrocatalytic bond formation under mild conditions. However, it is essential for the authors to clarify how their approach differs from and improves upon existing methods in the literature, specifically the work by Peiris et al. (Journal of the American Chemical Society 2019, 141, 14788-14797), which uses diazonium salts, and the studies published in Nanotechnology 2015 (26, 381001) involving terminal alkynes. Additionally, comparison should be made with covalent bonding strategies with silicon as seen in Nature Communications 2017 (8, 15056) and Chemical Science 2020 (11, 5246-5256).

While the authors highlight the robustness and potential advantages of their method, it remains unclear how these benefits are quantitatively or qualitatively more significant than those offered by the above-mentioned studies. A discussion that critically compares the efficiencies, stability, and practical applications of the Au-C bonds formed by their method versus the existing ones would significantly enhance the manuscript. I recommend that the authors include this comparative analysis to provide a clearer understanding of the true novelty and advantages of their approach.

Reviewer #4

(Remarks to the Author)

In the manuscript, Zhang, et al., report the formation of covalent Au-C bonds in single-molecule junction through electron-transfer-driven dissociation of Katritzky salts. They carried out a range of experiments on two different molecular series (PyBz-n and PhTMP-n), including 1) STM-BJ conductance measurements, 2) Flicker noise measurements, 3) Spectroelectrochemistry measurements, 4) EPR measurements, 5) Current rectification measurements on self-assembled monolayers (SAMs), and 6) DFT-based calculations. Despite their comprehensive study reported in their manuscript, there is no clear evidence of forming Au-C bonded junctions. We also have some major comments which we believe invalidate their conclusions making this work not acceptable for publication.

1. The authors claim to have a robust mechanism to create Au-C linked molecular junctions. However, there have been other methods reported in the literature, which they summarize in the supporting information document (Scheme 1) which show four other ways to create Au-C bond. Importantly, the fourth method (starting with halogen terminated molecules) seems far simpler than what is done here. So it is not clear that this work is sufficiently novel to merit publication in Nature communications. Why is it simpler to do what the authors do, rather than simply use halogenated compounds, which are often even commercial?

2. The main series that relates to the claim of this work is the one the authors call PhTMP-n. There are three molecules in this series and the authors claim that once the trimethylpyridine group leaves, they are left with a C-radical terminated molecule which should bind to Au electrodes with an SMe linker on one end and an Au-C linker on the other. The data for this series is presented in Figure 3 for PhTMP-1 and for the longer molecules in the SI. The measured conductance (10-2.9G0) is far too low for their proposed junction geometry (see Chen, et al., J. Am. Chem. Soc. 2011, 133, 17160-17163, the conductance of Au-CH2-Ph-CH2-Au with covalent Au-C bond on both sides is 0.9G0). This is also very low compared to a single Ph linked with two SMe linkers. Clearly, the authors are measuring something else here, and not an Au-C-Ph-SMe-Au junction. What is even more odd is that the conductance of PhTMP-1 is lower than Py-1 which has no Au-C contacts. The authors also claim a step length of 0.2 nm but it is not clear how this is determined. From Figure 3, it seems like the step length is closer to 0.4 nm indicating that they are measuring something longer. The color scheme of their 2D figures makes it really hard to figure out the lengths. More critically, the conductance of PhTMP-2 and PhTMP-3 are very close (0.6 e-3 and 0.12e-3 G0). Fitting this data gives a beta of 2.6/nm. This is too small to correspond to an oligophenylene series. For the measurement of PhTMP-1, it is plausible that the authors are actually measuring the dimer product of Ph-1 (MeS-Ph-CH2-CH2-Ph-SMe) without forming the Au-C bond on the Au electrodes. If this is indeed the case, then the entire premise of this work is incorrect.

3. In the flicker-noise measurement results for the PyBz-n series (Fig. 2e, and Supp. Fig. 21), the authors observed that the noise power scaling factor (N) decreases from 1.8 to 1.0 with increasing voltage bias. The manuscript attributes this change

to a transition from through-space tunneling in PyBz-3 to through-bond tunneling in Py-3. However, Supp. Fig. 21 and 22 show that the noise power data are within the same conductance range at different biases. This contradicts the fact that molecules with through-space coupling should exhibit much lower conductance than those with similar structures but with through-bond coupling (see Wei, et al., Nano Lett. 2023, 23, 567-572). Therefore, the entire flicker noise analysis does not support the authors' conclusions presented in the manuscript, as through-space coupling between pyridinium in PyBz-3 and gold should lead to a much lower conductance.

4. In the last section of the manuscript, the authors fabricated self-assembled monolayers of PhTMP-n on Au surfaces. The authors wrote: "The PhTMP-n molecules can only self-assemble on Au surface through the orderly and directionally alignment of the -SMe group." In the XPS results shown in Supp. Fig. 27c, the authors attribute the new peak at 288.7 eV to the formation of Au-C bonds. However, in Ref. 44 of this manuscript, the photoemission energy of Au-C bond is 282 eV, different from what they observe. Instead, the energy of 288.7 eV is likely to be a signal from C(sp³)-C(sp³) bonds (see Fujimoto, et al., Anal. Chem., 2016, 88, 6110-6114.), which also supports the dimerization of benzyl radicals.

Other points

1. The main claim of this paper is that they create Au-C linked molecular junctions. The authors however start with a presentation of junctions formed with pyridine and SMe linkers. This seems to be out of place. More importantly, it is only supporting information for their main claim since this set of molecules do not really show the formation of Au-C bonds.

2. The beta quoted in SI fig 17 is not correct. The authors seem to use plateau length since they claim that PyBz-1 has a length of 0.2 nm while PyBz-3 has a length of 1.3 nm. Also, they seem to be fitting the beta plot on a base-10 log scale, not a natural log scale which is again completely wrong.

3. The authors showed the flicker noise power results in Fig. 2e. We believe that the author should also provide the raw power spectral density before FFT. Also, did the authors select conductance traces for their flicker-noise data analysis? What are the selection criteria? How many traces are selected? This information should also be included in the manuscript.

4. In Supp. Fig. 24, the authors showed their CV data for PhTMP-n using glassy carbon as working electrode, Ag/AgCl as reference electrode, and Pt as counter electrode. For n = 1-3, the measured CVs exhibit irreversible reduction process at different potentials. However, these results do not necessarily reflect what happens in molecular junctions with gold electrodes, as gold has a different reactivity from glassy carbon. The authors should instead measure in-situ CVs using their STM-BJ setup.

Reviewer #5

(Remarks to the Author)

Version 1:

Reviewer comments:

Reviewer #1

(Remarks to the Author)

A pleasure to read and a significantly improved manuscript.

Only one question: why potassium tetrafluoroborate as supporting salt? It is very unusual to see this used as electrolyte in MeCN since it very limited solubility. This forced the authors to add a salt at 1 mM concentration where the Debye length will be significantly larger than the molecule hence poor fields near the electrode, leading to slow kinetics of electron transfer.

Reviewer #2

(Remarks to the Author)

The authors have carefully addressed my comments providing additional information that accurately presents the current state of the art in the field and additional experimental and theoretical proofs that support their findings. I strongly encourage the inclusion of figures R5, R6, R7 and the related explanations at least into the Supplementary Information. This material is essential and supports the interpretation of the results.

However, the inclusion of the Au/PhTMP-n/GaOx/EGaIn system in the main text, is still not convincing. These junctions do not have any relation to the main message of the manuscript that addresses the electrocatalytic Au-C bond formation.

Moreover, the envisaged rectification effect of the asymmetric coupling (Au-S vs. Au-C) of a molecular system should be demonstrated in junctions with similar electrodes not in asymmetric junctions with different electrodes. Besides, the experimental system Au/PhTMP-n/GaOx/EGaIn is different from the Au/Ph-n/Au considered for theoretical calculations. After addressing the final comments, I recommend the acceptance of the manuscript for publication.

Reviewer #3

(Remarks to the Author)

Authors reviewed the manuscript adequately. I recommend publication.

Reviewer #4

(Remarks to the Author)

Revised report:

The authors have attempted to address a few of the concerns raised however, from their revision, it is even more clear that they have not achieved their main claim, i.e. to create Au-C linked molecular junction starting with the pyridine salts. More importantly, they have simply not addressed many points in the review. Additionally, some new data added further weakens the novelty of this work.

1. As stated in the introduction of their manuscript, "The stability of Au-C covalent bonds in air and water, along with their unique electronic properties, make them highly advantageous for achieving [6-9] stronger coupling strength compared to other commonly used anchor groups such as amines^{10, 11}, thiols [12], thiomethyls^[13], pyridines [14, 15], etc." In their revision, they argue that the reason they observe a low conductance for the PhTMP series is because they have a methylene group which they claim reduces the conductance. The entire point of Au-C links is to have a high conductance. If they cannot achieve this, then this manuscript has limited novelty.

2. They claim that the low conductance observed in their junctions is Au-C terminated junctions by comparing the conductance of a new molecule SCh2-1 which has a lower conductance compared to S-1. However, this completely misses the point of using C-as the linker. When you create an Au-S-CH2-Ph linked system, there are two atoms between the phenyl and Au (S and CH2). However, for the Au-CH2-Ph, there is only one linker atom, the carbon. Their argument is therefore completely not valid. Secondly, In Chen et al, 133, 17160–17163 (2011), the conductance of the entire series is very high because there is a Au-CH2-Ph linker (their table in the SI – scheme 1 - is incorrect.) Without the CH2 group, the Au does not couple into the pi-system and conductances are lower. The bottom line is that these authors have not demonstrated the creation of Au-C linked molecular junctions.

3. The authors have not addressed why the beta value obtained for the PhTMP-n series is very low. This is a critical issue. To prove that you can measure a series with a new linker, the first step is to show that the series gives a correct conductance decay. This has not been done. They also do not provide a clean comparison of conductance vs length for this series.

4. The new x-ray data added shows carbon peaks even on clean gold – this is a major problem. They do not have clean Au surfaces and thus the rest of their data cannot simply be attributed to the system they study.

5. The authors have not shown any FFT data in the manuscript or SI.

6. If the authors indeed have a new way to create Au-C bonded junctions, they should demonstrate this with both sides of the molecule linked with Au-C bonds. With their method, this is not possible however with halide terminations, this is trivial.

7. The authors claim that the problem with halide terminations is that it can lead to dimerization. This is correct but in their system, they should also see dimerization. Any method that creates a benzyl radical as an intermediate has the possibility to dimerize. This is the case with halides, SnMe₃ and any other pathway. The fact that they now claim they do not see dimerization indicates that they do not create the radical species as an intermediate.

Reviewer #5

(Remarks to the Author)

Version 2:

Reviewer comments:

Reviewer #2

(Remarks to the Author)

The authors have carefully revised the manuscript, removing the unrelated information on self-assembled monolayers and including additional experimental results related to the formation of covalent Au-C contacts at both molecular termini. These results demonstrate a significant enhancement in conductance.

Regarding the new data, I recommend specifying the applied bias voltage for the different measurements presented in

Figure 4.

As a minor suggestion to support visual interpretation and facilitate differentiation between the presented systems, I propose reversing the displacement axis in Figure 4b (i.e., displaying decreasing displacement values). This would reflect the actual measurement procedure, which involved approaching the tip toward the surface ("tip-surface closing process"), in contrast to the other systems discussed in the manuscript.

Once the final comments are addressed, I believe the manuscript is suitable for publication.

Reviewer #4

(Remarks to the Author)

I appreciate the authors for taking my comments seriously and dedicating time to address them. Their PhSn-n control experiments convincingly demonstrate the formation of molecular junctions with Au-C covalent bonds. Furthermore, they showed that they were able to form Au-C bonds with both electrodes using Ph(TMP)₂-n, leading to an exponential decay in great agreement with previous findings, which reinforces the significance and novelty of this work.

According to the manuscript, SET is the rate-limiting step of Au-C formation, which implies the steady-state concentration of benzyl radical is very low, thereby reducing the likelihood of radical dimerization. However, it would still be nice to include another control experiment using a radical scavenger (such as butylated hydroxytoluene). If the presence of a scavenger inhibits or blocks the Au-C junction formation, it would be helpful on diagnosing the mechanism. Beyond this point, I have some minor comments that should be addressed:

1. The authors show some noise analysis (Figure 2e) however there are no details on how this noise is calculated. Either the authors should show raw data that demonstrates how they make these images or simply remove this panel from the figure as it does not really add any new information.
2. The claim in the abstract that the conductance of the 8A wire is 1 Go is a bit misleading. Figure 4e shows a dark black fit over the data which hides the raw data behind it. From the 2D histogram in Figure 4b, it is clear that the peak is below 0. They should not overstate the findings with the statement "a record value for molecular wires of comparable length" since this value is basically the same as found by earlier researchers.
3. The 2D histograms in Figure 4 all show some vertical lines. Is this a resolution problem?

Reviewer #5

(Remarks to the Author)

Version 3:

Reviewer comments:

Reviewer #4

(Remarks to the Author)

The authors have addressed all concerns raised.

Listed below are the details of our responses to the referees' comments.

Reviewer #1

Since their discovery in the 1970s, Katritzky salts have become one of the most significant classes of building blocks in organic synthesis. These bulky pyridinium salts can participate in virtually any known reaction, alkylation, arylation, alkenylation, alkynylation, carbonylation, sulfonylation, and borylation. The current report expands their use to molecular electronics, specifically in the context of forming Au-C bonds. It should definitely feature in a good quality journal like Nat Commun.

Reply

We express our gratitude to this esteemed reviewer for the favorable assessment of our research. We are honored by the recommendation to publish our work in a high-profile journal like *Nature Communications*.

- *1. One oversight by the authors of this technically very sound paper is that there is no mention of the very first paper describing how to make Au-C based molecular junctions. It was in a not-so-visible journal, but nonetheless, since it was the first, it should be cited and discussed (Pla-Vilanova, P.; Aragonès, A. C.; Ciampi, S.; Sanz, F.; Darwish, N.; Diez-Perez, I., The spontaneous formation of single-molecule junctions via terminal alkynes. *Nanotechnology* 2015, 26, 381001).*

Reply

Thanks for this reviewer for bringing to our attention the omission of the seminal paper on Au-C based molecular junctions by Pla-Vilanova et al. We sincerely apologize for this oversight. The work is indeed the very first to report the spontaneous formation of single-molecule junctions via terminal alkynes, and it plays a foundational role in the

development of Au–C based molecular junctions. We have now cited and discussed this paper in the introduction section of our manuscript, highlighting its relevance and how our study builds upon their pioneering work. We believe this paper provides important context and further strengthens the significance of our findings.

We have added this reference together some other methods developed to create Au–C in single-molecule junctions on **Page 2 and 3** in the main text.

“...To date, many methods (Supplementary Scheme 1) have been developed to create an Au–C bonded interface. Firstly, terminal alkynes can form the Au–C bonds spontaneously¹⁷, with neither deprotonation agents nor external stimuli...”

The comparison of different methods to form Au–C bonds in molecular junctions have also been summarized in **Scheme 1** and **Table 1** on **Page S4 and S5** in the SI.

Table 1. The comparison of the efficiency and stability of Au-C bonds formed by existing method.

Method	Stability	Efficiency	Advantages	Disadvantages	Reference
1	★ ★ ★	★ ★	In-situ	Highly toxic substrate	1
2	★ ★ ★	★ ★	In-situ	Highly toxic substrate	2
3	★ ★ ★	★ ★ ★	Spontaneously	Unstable substrate	3
4	★ ★ ★	★ ★ ★	Stably and efficiently	Ex-situ	4
5	★ ★ ★	★ ★ ★	Environmentally friendly	Unstable substrate	5–6
6	★ ★	★ ★ ★	High efficiency	Unstable substrate Prone to side reactions	7
7	★ ★ ★	★ ★ ★	Environmentally friendly	Prone to coupling reaction	8
8	★ ★ ★	★ ★ ★	Environmentally friendly	Limited substrate	9
9	★ ★ ★	★ ★	Stably and efficiently	Ex-situ	10
10	★ ★ ★	★ ★ ★	Stably and efficiently	Highly toxic substrate	11
Si–C bond	★ ★ ★	★ ★ ★	Stably	Limited substrate	12–13

Figure R1. The methodologies developed to create Au–C covalent bonding at the interface.

- 2. *The paper is well researched and presented, and my only remark is on the possible involvement of the solvent. Is it possible that the reaction is indeed spontaneous (or become even more so) if a system that favors electron transfer and screens Coulombic interactions between surface holes and electrons is selected. I am aware this is not a semiconducting substrate, but e/h pairs form transiently even in metals, and it is possible that the time scale of the surface reaction is comparable to the lifetime of e/h pairs in Au. A discussion on how to test for this “hidden” spontaneous (electrochemical in nature) surface reactions was described by J. Gooding about 10 years ago in Phys. Chem. Chem. Phys. 2014, 16, 8003.*

Reply

We appreciate this reviewer’s suggestion to consider the potential role of solvent effects and the electrochemical nature of the reaction in our study. Although our primary focus was on the formation of Au–C bonds via single electron transfer (SET) from Au electrodes to pyridinium salts, we accept the fact that the solvent environment could indeed influence the reaction, especially by affecting (*J. Chem. Phys.* **1989**, *90*, 1720–1729; *Phys. Chem. Chem. Phys.* **2014**, *16*, 8003) electron transfer dynamics and Coulombic interactions. We have performed numerous control experiments to assess the impact of solvents.

Firstly, the formation of Au–C bonds between Au electrodes and pyridinium salts via single electron transfer (SET) cannot occur spontaneously. Taking **PyBz-3** as an example, individual breaking curves obtained from 0.1 to 0.7 V are presented in **Figure R2**. It is evident from the figure that there is no significant conductance signal from 0.1 to 0.3 V. As the bias voltages increase gradually, the conductance signal of the **PyBz-3** at around $10^{-4.0} G_0$ becomes increasingly apparent, as also can be seen in

Supplementary Information Figure 17. A step with a length of 0.78 nm emerges gradually after the bias voltage reaches 0.4 V, consistent with the signal of **Py-3** synthesized ex situ. Therefore, we conclude that the single electron transfer (SET) reaction is initiated by the electrons injected from the needle tip as the bias voltages increase to a certain threshold level.

Figure R2. Bias-dependent conductance measurements of PyBz-3 (a–g) The single traces were obtained at various bias voltages of 0.1 (a), 0.2 (b), 0.3 (c), 0.4 (d), 0.5 (e), 0.6 (f), and 0.7 V (g).

Secondly, we conducted a cyclic voltammetry test using an Au electrode as the working electrode. As shown in **Figure R3**, the reduction potential of **PhTMP-1** is approximately -0.9 V. In the scanning range from 0 to -0.6 V, the current remains at a minimal level, almost reaching zero. Beyond -0.62 V, the current exhibits a sudden increase, indicating that the reaction is non-spontaneous and necessitates bias voltages to facilitate electron injection and drive the reaction conditions.

Figure R3. Cyclic voltammograms (CVs) experiments performed on PhTMP-1. The CV experiments were performed in MeCN with potassium tetrafluoroborate (KBF_4 , 1 mM) as the auxiliary electrolyte. Working electrode (WE): Au; Reference electrode (RE): Ag/AgCl; Counter electrode (CE): Pt.

Figure R4. Bias-dependent conductance measurements of PyBz-3 using propylene carbonate (PC) as the solvent. **a**, Scheme illustration showing the SET-induced transformation from **PyBz-3** to **Py-3** at increasing bias voltages starting from **b**, 0.1 V, **c**, 0.2 V, **d**, 0.3 V, **e**, 0.4 V, **f**, 0.5 V to **g**, 0.6 V. A conductance peak appeared gradually at around $10^{-4.0} G_0$ at a bias of 0.3 V, and its prominence increased as the bias voltage was raised up to 0.6 V.

Finally, we investigated the effect of solvent polarity on the reaction and found that while increased solvent polarity can promote the reaction to some extent, it is insufficient to drive the reaction spontaneously. To evaluate the impact of solvent polarity, we tested solvents with varying polarities. However, as demonstrated by **PyBz-3**, pyridinium salts have limited solubility in non-polar solvents such as TCB, which is commonly used in our experiments. Consequently, we conducted STM-BJ experiments in a more polar solvent, propylene carbonate (PC). The results, shown in **Figure R4**, indicate that no significant conductivity was observed at 0.1 and 0.2 V in PC. A conductance peak appeared gradually at approximately $10^{-4.0} G_0$ at a bias voltage of 0.3 V, suggesting the occurrence of SET-induced homolysis of the C–N bond in **PyBz-3**. This voltage is lower than the 0.4 V observed in MeCN, suggesting that the more polar solvent may enhance electron transfer, even though the reaction remains non-spontaneous.

Reviewer #2

The manuscript “Electrocatalytic Au–C bond formation in single-molecule devices” by Yaxuan Zhang et al. describes the charge transport through single molecules using a scanning tunneling microscope-based break junction (STM-BJ) technique. The authors report that the injection of electrons by STM-BJ into molecular systems consisting of N-functionalized Katrizky salts, lead simultaneously, to the scission of a C-N chemical bond releasing pyridines molecules and benzyl radicals that form covalent Au–C bonds with Au electrodes. The strong molecule-to-electrode anchoring and the understanding of the charge transport at the individual molecular level has significant appeal for the progress of single-molecule-based electronic devices.

Reply

We appreciate this reviewer’s summary and recognition of the significance of our work in understanding charge transport at the molecular level and its implications for single-molecule electronics. The remarks raised by this reviewer have reinforced the relevance of our study to the field, and we are encouraged by this reviewer’s support for its publication.

- *1. The authors mention in the introduction and Supplementary Information (SI) that up to date only four methods have been developed to form covalent Au-C bonds. Regarding this aspect, the literature review is outdated. The methodologies for forming C-metal covalent bonds have moved on recently. The description of this field should be refreshed.*

Reply

We appreciate this reviewer’s suggestion and agree that our review of the field should reflect the latest advancements. We have updated the Introduction and Supplementary Information (SI) with recent literature. The comparison of different methods to form

Au–C bonds in molecular junctions have also be summarized in **Scheme 1** and **Table 1** on **Page S4 and S5** in the SI. The additional information includes new methodologies that have emerged in the formation of covalent Au–C bonds. This ensures our manuscript accurately represents the current state of the field.

- *2. The authors claim that the dissociation of the C-N bonds of the presented molecules occurs during the conductance measurements by STM-BJ. All the conductance curves presented in the manuscript and the SI show no clear evidence of such reactions. STM techniques are very sensitive to any molecular changes in the junction, even as geometrical conformations. The conductance traces and the peak visible in the conductance histograms prominent at certain bias voltages, have a standard appearance for the conductance through molecular junctions. The negligible conductance features at bias voltages close to the Fermi level may be related to the lack of molecular states at the corresponding energies.*

Reply

We thank this reviewer for the valuable comments regarding conductance changes at different bias voltages. **Firstly**, as depicted in the mechanism diagram of **Scheme 2**, the homolysis reaction occurs rapidly after electron injection into the molecule. Therefore, we believe that only the pre-reaction and post-reaction states can be measured using STM-BJ. However, detecting the signal of pyridinium salt compounds before the reaction is challenging because these molecules rely solely on electrostatic adsorption to the substrate surface. After the reaction, only **Py–n** or **Ph–n**, which establish covalent (Au–C) or dative (Au–N) contacts with Au, can be monitored. The detection of the other homolysis product is challenging due to the lack of an anchoring group. While the STM technique is exceptionally sensitive to molecular changes in the junction, encompassing geometric conformations, the unique characteristics of our

reaction lead to the lack of detectable intermediate species. Our measurements of the “post-reaction” species demonstrate conductance corresponding to **Py-n** or **Ph-n**, which is a typical observation within this specific context.

Secondly, according to our experimental findings, the conductance of **Py-n** is assessed following the application of a high bias voltage. Upon conducting the bias dependence experiment on **Py-n**, it was noted that its conductance remains constant regardless of variations in the bias voltage. This observation suggests that altering the bias voltage does not result in the alignment of molecular energy levels with the Fermi level. Using **Py-3** as an example, we collected the conductance signal of this molecule from 0.1 to 0.7 V, and presented its 2D conductance-displacement histograms in **Figure R5**. The conductance characteristics of **Py-3** do not change (**Figures R5** and **R6**) significantly with increasing bias voltages.

Finally, we employed electron paramagnetic resonance (EPR) and UV-Vis spectroelectrochemical ensemble experiments, which indirectly confirm the feasibility of the reaction.

Figure R5. Bias-dependent conductance measurement of Py-3 (a-g) The 2D conductance-displacement histograms at various bias voltages of 0.1 (a), 0.2 (b), 0.3 (c), 0.4 (d), 0.5 (e), 0.6 (f) and 0.7 V (g).

Figure R6. Bias dependent experimental conductance statistics of Py-3.

- 3. *The similarities of the conductance features of PyBz-n (that are supposed to become Py-n) and the ex-situ prepared Py-n molecules should appear at similar bias voltages. The observed difference, that may be related to a different electronic structure, supports the previous comment. The PyBz-n and PhTMP molecules may have junction conformations similar to the ones before the reaction sketched in the left panels of the Supplementary Figures 14(a), 15(a), 16(a), 18(a), 19(a), 20(a). Thus, the charge transport may occur mostly through Py-n and Ph-n moieties where the orbitals that dominate the transport may be located, which may not have any weight on the Bz and TMP groups. Examples of similar scenarios can be found in the literature.*

Reply

We appreciate this reviewer's suggestion to consider the conductance characteristic of ex-situ **Py-n**.

Firstly, we synthesized **PhBz-2**, which shares a similar molecular skeleton as **PyBz-2**, with only one difference being the lack of the positively charged pyridinium group. Different from the situation in **PyBz-2**, no obvious conductance change was

detected (**Figure R7**) in the control compound **PhBz-2**, even at higher bias voltages. As a result, it is far to draw the conclusion as this reviewer mentioned that “...*the charge transport may occur mostly through Py-n and Ph-n moieties where the orbitals that dominate the transport may be located...*”

Our observation provides compelling evidence that the presence of the positively charged pyridinium unit is essential for the initiation of the single electron transfer (SET) reactions. The electrostatic binding junctions of **PhBz-n** or **PhTMP-n** are challenging to detect at low bias voltages. Only **Py-n** or **Ph-n** formed after the homolytic reaction from **PhBz-n** or **PhTMP-n** at high bias voltages can be identified.

Figure R7. Bias-dependent conductance measurement of PhBz-2 (a-f) The 1D and 2D conductance-displacement histograms at different bias voltages starting from 0.1 V (a), 0.2 V (b), 0.3 V (c), 0.4 V (d), 0.5 V (e) to 0.6 V (f).

Secondly, analysis of the transmission spectra (**Figure R8**) calculated for **PyBz-3** revealed that the conductance of **PyBz-n** with electrostatic anchors is remarkably dependent on the distance between the molecule and the Au electrode. At a distance of 9 Å (**Figure R8a**), the theoretical conductance (ranging from 10^{-8} to 10^{-12}) is considerably lower than the values observed in experimental studies. This finding demonstrates that the formation of Au–N or Au–C bonds is a crucial step for achieving conductance at high bias voltages.

We have added the relevant information in **Figure R5**, and in **Figure 2** on **Page 7** and **8** of the main text, we have updated the conductance of **Py-3** to the value measured at 0.7 V.

Figure R8. DFT calculated transmission spectra for the junctions PyBz-3 (a) The optimized junction geometries for **PyBz-3**. (b) The calculated transmission functions for **PyBz-3**. The distance between the gold electrode and methylene is changed, and the calculated transmission spectrum is shown in **c, d**.

- 4. *This manuscript focuses on the formation of covalent Au-C bonds. It is not clear why the authors present the Au/PhTMP-n/GaOx/EGaIn monolayer junctions in the last part of the manuscript, that is not directly related to the main focus of the manuscript and the description is overreached with links to SI. It gives the impression that the corresponding part is introduced to expand the length of the manuscript.*

Reply

We sincerely thank the reviewer for their valuable comments. Our work focuses on two key aspects: (1) the primary objective of our article is the development of a methodology for electrocatalytically forming Au–C bonds, which is central to our research, and (2) the use of asymmetric coupling (Au–S vs. Au–C) to construct molecular rectifiers, which is one of the techniques currently reported for building such devices (*Nano Lett.* **2013**, *13*, 6233–6237). The outcomes from the STM-BJ experiments validate the rectification effect, albeit with a relatively low rectification ratio. We are dedicated to pursuing additional research in this field.

What sets our system apart from others is the asymmetric structure of the molecule prior to the reaction, showcasing exceptional assembly traits. The –SMe anchor forms a more robust contact with the Au electrode, whereas the positively charged pyridinium unit binds with lesser strength. The existence of asymmetric anchors enables thermodynamic control during solution assembly, wherein the –SMe end selectively interacts with the Au surface, leading to the formation of a monolayer with a specific orientation. This design forms the basis for the subsequent development of large-scale rectifying monolayer devices. Initial findings from the EGaIn experiments present in **Figure 4** further confirmed this design.

The EGaIn experiment, serving as a unique characterization method distinct from STM-BJ, additionally illustrates that this approach of SET-induced reaction is predominantly influenced by the electrode interface and is not restricted to particular electrode materials. This adaptability underscores the wider applicability of our approach.

Moreover, the experiment furnishes compelling evidence that our strategy can be utilized successfully in the creation of large-area functional devices. This study exemplifies the crucial linkage between laboratory research and practical applications. It signifies a substantial advancement in translating experimental discoveries into tangible technologies.

Reviewer #3

The authors present an approach for forming covalent Au–C bonds in single-molecule devices through single electron transfer (SET) from gold electrodes to pyridinium salts. This methodology introduces some elements of novelty, particularly in the context of electrocatalytic bond formation under mild conditions.

Reply

Thanks very much for this reviewer's thoughtful review and for acknowledging the novelty of our work, particularly the methodology involving single electron transfer (SET) to form Au–C covalent bonds under mild conditions in single-molecule devices. We greatly appreciate your positive comments regarding the electrocatalytic bond formation approach.

- *1. However, it is essential for the authors to clarify how their approach differs from and improves upon existing methods in the literature, specifically the work by Peiris et al. (Journal of the American Chemical Society 2019,141, 14788-14797), which uses diazonium salts, and the studies published in Nanotechnology 2015 (26, 381001) involving terminal alkynes. Additionally, comparison should be made with covalent bonding strategies with silicon as seen in Nature Communications 2017 (8, 15056) and Chemical Science 2020 (11, 5246-5256).*

Reply

We have expanded our discussion to clearly outline the novelty of our method, particularly in the context of electrocatalytic bond formation under mild conditions. Specifically, we discussed their differences as follows.

Firstly, diazonium halides in solid-state are frequently hazardous, as they can turn explosive when dry, leading to reports of fatalities and injuries linked to their utilization. This significantly limits their widespread application, which poses challenges for the future fabrication of electronic devices. Although effective, the formation of diazonium salts often require harsh conditions, such as strong reducing agents. In contrast, our method employs pyridinium salts, which form Au–C bonds through single-electron transfer (SET) under more mild electrochemical conditions. These mild conditions not only reduce the likelihood of side reactions, but also introduce an electrocatalytic approach where the electrode participates in bond formation via SET, offering a potentially more efficient route.

Secondly, research on terminal acetylenes has primarily focused on the formation of Au–C bonds from terminal alkyne (*Nanotechnology* **2015**, *26*, 381001). One advantage of this approach is the high reactivity of the terminal alkyne. The alkyne structure also exhibits a certain degree of rigidity similar to that of OPE, which leads to distinct conductance properties and makes it an excellent molecular backbone for electronics.

However, the terminal alkyne hydrogen (H) is highly reactive and typically requires protection by a trimethylsilyl (TMS) group. These methods rely on strong bases (such as TBAF) to remove the TMS protecting group, which is not an in situ formation of Au–C bonds. Instead, our method relies on pyridinium salts to generate carbon radicals in situ and subsequently form Au–C bonds directly with the Au electrode. This method simplifies the preparation process and opens up the possibility of utilizing more molecules without requiring alkynyl groups.

Thirdly, silicon substrates are extensively utilized in the microelectronic industry, and research in this field has investigated numerous covalent bonding strategies to create robust silicon-carbon (Si–C) bonds. These efforts are expected to build a bridge between molecular electronics and silicon-based microelectronics in the future. While silicon-based methods are effective, it is equally crucial to establish dependable covalent contacts on various materials  such as metal electrodes and metal oxide dielectric materials  in electronic devices. In this context, investigating Au–C bonds is advantageous for exploring strong covalent connections within inert metal electrode materials.

More detailed contents are summarized in **Table R1**.

- *2. While the authors highlight the robustness and potential advantages of their method, it remains unclear how these benefits are quantitatively or qualitatively more significant than those offered by the above-mentioned studies. A discussion that critically compares the efficiencies, stability, and practical applications of the Au–C bonds formed by their method versus the existing ones would significantly enhance the manuscript. I recommend that the authors include this comparative analysis to provide a clearer understanding of the true novelty and advantages of their approach.*

Reply

We appreciate this reviewer's suggestion and have added a comparative analysis to the manuscript.

Firstly, we analyzed the efficiency of the SET homolytic reaction using **PyBz-3** as an example. The results are shown in **Figure R9**. It is observed that as the bias voltage increases to 0.4 V, the yield of the homolytic reaction sharply rises to 60%. While we anticipate that higher yields could be attained at higher bias voltages, additional data collection is constrained by instrument limitations.

Figure R9. Distribution probabilities of homolysis for PyBz-3 plotted against different biases.

Table R1. The comparison of the efficiency and stability of Au-C bonds formed by existing method.

Method	Stability	Efficiency	Advantages	Disadvantages	Reference
1	★ ★ ★	★ ★	In-situ	Highly toxic substrate	1
2	★ ★ ★	★ ★	In-situ	Highly toxic substrate	2
3	★ ★ ★	★ ★ ★	Spontaneously	Unstable substrate	3
4	★ ★ ★	★ ★ ★	Stably and efficiently	Ex-situ	4
5	★ ★ ★	★ ★ ★	Environmentally friendly	Unstable substrate	5-6
6	★ ★	★ ★ ★	High efficiency	Unstable substrate Prone to side reactions	7
7	★ ★ ★	★ ★ ★	Environmentally friendly	Prone to coupling reaction	8
8	★ ★ ★	★ ★ ★	Environmentally friendly	Limited substrate	9
9	★ ★ ★	★ ★	Stably and efficiently	Ex-situ	10
10	★ ★ ★	★ ★ ★	Stably and efficiently	Highly toxic substrate	11
Si-C bond	★ ★ ★	★ ★ ★	Stably	Limited substrate	12-13

Additionally, the existing methods for forming Au–C covalent bonds are listed in **Table R1** and **Figure R10**. Compared to these methods, we demonstrate that our electrocatalytic approach offers superior control over bond formation under mild conditions, which can lead to more stable molecular junctions. Additionally, we discuss the practical applications, highlighting our method's potential for scalability and integration into device fabrication.

Figure R10. The methodologies have been developed to create Au–C covalent bonding at the interface.

Reviewer #4

In the manuscript, Zhang, et al., report the formation of covalent Au-C bonds in single-molecule junction through electron-transfer-driven dissociation of Katritzky salts. They carried out a range of experiments on two different molecular series (PyBz-n and PhTMP-n), including 1) STM-BJ conductance measurements, 2) Flicker noise measurements, 3) Spectroelectrochemistry measurements, 4) EPR measurements, 5) Current rectification measurements on self-assembled monolayers (SAMs), and 6) DFT-based calculations.

Reply

We thank this reviewer for the positive comments on the novelty and scientific relevance of our manuscript. As for this reviewer's questions and concerns, we have done our best to provide appropriate, reasoned and thoughtful responses to them.

- *1. The authors claim to have a robust mechanism to create Au-C linked molecular junctions. However, there have been other methods reported in the literature, which they summarize in the supporting information document (Scheme 1) which show four other ways to create Au-C bond. Importantly, the fourth method (starting with halogen terminated molecules) seems far simpler than what is done here. So it is not clear that this work is sufficiently novel to merit publication in Nature communications. Why is it simpler to do what the authors do, rather than simply use halogenated compounds, which are often even commercial?*

Reply

We thank this Reviewer for pointing out this issue. Firstly, one challenge is reducing the possibility of electric-field-catalyzed dimerization of aryl iodides. Stone

et al. (*Chem. Sci.* **2022**, *13*, 10798–10805) reported that aryl iodide compounds efficiently undergo Ullmann coupling reactions on rough Au surfaces during STM-BJ experiments. They proposed that interfacial electric fields contribute to catalyzing the Ullmann reaction. However, in our positively charged pyridinium compounds, the Coulombic repulsive forces provided by the positive charges, along with the steric protection and leaving groups, significantly reduce the dimerization of carbon radicals.

Secondly, it is challenging to quantitatively determine the exact voltage required for the formation of Au–C bonds in aryl iodides. Halogenated aromatic compounds are electrochemically inactive, so Starr et al. (*J. Am. Chem. Soc.* **2020**, *142*, 7128–7133) qualitatively reported that aryl iodides form Au–C interactions under negative tip bias and Au–I interactions under positive tip bias. However, in our study, pyridinium salts have been demonstrated (*J. Org. Chem.* **2019**, *84*, 8759–8765; *Nature* **2023**, *623*, 949–955) to exhibit prominent redox peaks in the range of –0.9 to –1.4 V when using Ag/AgCl as a reference electrode. Consequently, the bias voltages applied to initiate the SET reaction are associated with the reduction potential of the pyridinium salts under investigation. By further optimizing the testing conditions and the electrocatalytic environment, we anticipate achieving semi-quantitative or even quantitative control over the voltage required for Au–C bond formation in the future.

➤ *2. The main series that relates to the claim of this work is the one the authors call PhTMP-n. There are three molecules in this series and the authors claim that once the trimethylpyridine group leaves, they are left with a C-radical terminated molecule which should bind to Au electrodes with an SMe linker on one end and an Au-C linker on the other. The data for this series is presented in Figure 3 for PhTMP-1 and for the longer molecules in the SI. the measured conductance (10-2.9G0) is far too low for their proposed junction geometry (see*

Chen, et al., J. Am. Chem. Soc. 2011, 133, 17160-17163, the conductance of Au-CH₂-Ph-CH₂-Au with covalent Au-C bond on both sides is 0.9G₀). This is also very low compared to a single Ph linked with two SMe linkers. Clearly, the authors are measuring something else here, and not an Au-C-Ph-SMe-Au junction.

Reply

We appreciate this reviewer's comments, and we believe that the presence of methylenes (-CH₂-) reduces conductance of **Ph-n•** backbones.

Firstly, we summarized the single-molecule conductance signals of 1,4-benzenedithiol (1,4-BDT) based on recent data. As shown in the **Figure R11**, there is a three-order-of-magnitude difference in the conductance of 1,4-BDT, which we attribute the variations to the STM-BJ set-up and the test environment.

Figure R11. 1,4-BDT conductance summary in the past 20 years.

Secondly, we synthesized (**Figure R12**) the control compounds with various methylenes (-CH₂-) linkers, i.e., **S-1**, **SCH₂-1**, and **S(CH₂)₂-1** to perform conductance measurements using STM-BJ. We found that **S-1** demonstrated a high-conductance (HC) state at $\sim 10^{-2.0} G_0$, and a low-conductance (LC) state at $\sim 10^{-4.0} G_0$, respectively. According to previous work, the HC state can be attributed the monomer

of **S-1**, while the LC state can be attributed to the π - π dimerization of **S-1**. Similarly, the conductance of **SCH₂-1** at around $10^{-2.9} G_0$ can be attributed to the monomer, while the lower conductance around $10^{-4.2} G_0$ can be attributed to the π - π dimerization of **SCH₂-1**. In summary, **SCH₂-1** and **Ph-1** are similar in length and structure (with one -CH₂- linker), which explains their comparable conductance values.

Additionally, we compared our measurements of **S(CH₂)₂-1** with the data reported by H. Vázquez et al. (*Nat. Nanotechnol.* **2012**, 7, 663–667). The conductance of **S(CH₂)₂-1** from our measurement is around $10^{-3.50} G_0$, which is very close to the value ($10^{-3.45} G_0$) reported by H. Vázquez et al. In summary, we believe the observed signal corresponds to the Au-CH₂-Ph-CH₂-Au molecular junction.

Figure R12. a, The one-dimensional (1D) and two-dimensional (2D) conductance-

displacement histograms of molecule **S-1**. **b**, The one-dimensional (1D) and two-dimensional (2D) conductance-displacement histograms of molecule **SCH₂-1**. **c**, The one-dimensional (1D) and two-dimensional (2D) conductance-displacement histograms of molecule **S(CH₂)₂-1**.

- *3. What is even more odd is that the conductance of PhTMP-1 is lower than Py-1 which has no Au-C contacts. The authors also claim a step length of 0.2 nm but it is not clear how this is determined. From Figure 3, it seems like the step length is closer to 0.4 nm indicating that they are measuring something longer. The color scheme of their 2D figures makes it really hard to figure out the lengths. More critically, the conductance of PhTMP-2 and PhTMP-3 are very close (0.6×10^{-3} and 0.12×10^{-3} G₀). Fitting this data gives a beta of 2.6/nm. This is too small to correspond to an oligophenylene series. For the measurement of PhTMP-1, it is plausible that the authors are actually measuring the dimer product of Ph-1· (MeS-Ph-CH₂-CH₂-Ph-SMe) without forming the Au-C bond on the Au electrodes. If this is indeed the case, then the entire premise of this work is incorrect.*

Reply

We appreciate the reviewer's comments. In **Figure 3** on page 11 of our main text, the colorbar of conductance has been modified in order to see the length of conductance step more clearly. The area with the highest probability of breaking trace occurrence is 0.2 nm

In order to eliminate the possibility of homocoupling, we synthesized **C2** (**Figure R13**). The length of this molecule is 1.3 nm, which is significantly greater than the measured length of **Ph-1**. With two -CH₂- linkers in the molecular backbone, the

conductance of **C2** is decreasing to $10^{-4.7} G_0$, which is substantially lower than the conductance of **Ph-1** observed in our measurements.

Figure R13. The one-dimensional (1D) and two-dimensional (2D) conductance-displacement histograms of molecule **C2**.

We have added the supplementary conductance information on **Page S28** of the SI (**Supplementary Figure S22**) and included a description of the control molecule on **Page 12** of the main text:

“...Additionally, we synthesized the dimer control molecule *C2*, and the result revealed its conductance to be around $10^{-4.7} G_0$, which rules out the possibility of forming coupling by-products (Supplementary Fig. 22)....”

- 4. In the flicker-noise measurement results for the PyBz-*n* series (Fig. 2e, and Supp. Fig. 21), the authors observed that the noise power scaling factor (*N*) decreases from 1.8 to 1.0 with increasing voltage bias. The manuscript attributes this change to a transition from through-space tunneling in PyBz-3 to through-bond tunneling in Py-3. However, Supp. Fig. 21 and 22 show that the noise power data are within the same conductance range at different biases.

This contradicts the fact that molecules with through-space coupling should exhibit much lower conductance than those with similar structures but with through-bond coupling (see Wei, et al., Nano Lett. 2023, 23, 567-572). Therefore, the entire flicker noise analysis does not support the authors' conclusions presented in the manuscript, as through-space coupling between pyridinium in PyBz-3 and gold should lead to a much lower conductance.

Reply

We thank this reviewer's for raising this issue. Firstly, at low bias voltages, **PyBz-n** shows no significant conductance in the STM-BJ test due to the weak coupling of the pyridinium group to the Au electrode at one end, which is reflected as a slope in the 2D conductance histogram. As a result, in the hover experiment, we use the distinct conductance plateau around $10^{-4.0} G_0$, observed after the reaction at high bias voltages, as the feedback signal rather than distance. It is possible that at high bias, the molecular junction is fully stretched, whereas at low bias, the molecule is weakly coupled to the electrode via the positively charged pyridinium anchor. In this case, the molecule may lie flat on the substrate, with a shorter tip-substrate displacement required to reach the feedback conductance of $\sim 10^{-4.0} G_0$. Therefore, this observation is not inconsistent with the reviewer's reference that through-space coupling would result in much lower conductance.

- *5. In the last section of the manuscript, the authors fabricated self-assembled monolayers of PhTMP-n on Au surfaces. The authors wrote: "The PhTMP-n molecules can only self-assemble on Au surface through the orderly and directionally alignment of the -SMe group." In the XPS results shown in Supp. Fig. 27c, the authors attribute the new peak at 288.7 eV to the formation of Au-C bonds. However, in Ref. 44 of this manuscript, the photoemission energy of*

Au-C bond is 282 eV, different from what they observe. Instead, the energy of 288.7 eV is likely to be a signal from C(sp³)-C(sp³) bonds (see Fujimoto, et al., Anal. Chem., 2016, 88, 6110-6114.), which also supports the dimerization of benzyl radicals.

Reply

Thank you for your input on the XPS peak assignment. We apologize if there was any confusion regarding the literature cited in the article, as the carbon atom in TiC is not considered organic carbon. In the literature suggested by the reviewer, the binding energy of sp³-hybridized carbon is only 285.0 eV, and the peak at 288.7 eV is likely attributed to a highly electronegative element.

In order to clarify the attribution of the C species, we conducted (**Figure R14**) additional XPS analysis on control compounds. We found that both the pure Au substrate (**Figure R14 a**) and the control molecule **S-1** (**Figure R14 g**) exhibited a peak at 288.7 eV, indicating that this peak is most likely due to atmospheric carbon contamination. We have made the corresponding correction in the Supplementary Information (SI). The revised content has been included from **Page S34** to **S36** of the Supplementary Information (SI).

However, we observed that, compared to the bare Au substrate, the Au 4f peak of the monolayer formed by the self-assembly of **PhTMP-1** is chemically shifted by 0.4 eV, indicating the successful formation of the monolayer. Additionally, we included the XPS spectra of **S-1**, where the Au 4f peak shows a 0.1 eV shift, further supporting our conclusions.

Figure R14. X-ray photoelectron spectroscopy (XPS) characterizations showing the Au 4f distinction of **b**, the PhTMP-1 powders, **f**, PhTMP-1 SAMs and **g**, S-1 SAMs.

Other points

- 6. *The main claim of this paper is that they create Au-C linked molecular junctions. The authors however start with a presentation of junctions formed with pyridine and SMe linkers. This seems to be out of place. More importantly, it is only supporting information for their main claim since this set of molecules do not really show the formation of Au-C bonds.*

Reply

We thank this Reviewer for raising this issue. Actually, we initially explored the compound **PyBz-*n*** to verify the feasibility of the single-electron transfer (SET) mechanism in STM-BJ experiments. In our study, **PyBz-*n*** received electrons injected by the STM tip, underwent a SET-induced homolytic reaction, and simultaneously formed **Py-*n*** compounds. These newly formed compounds featured a sulfur methyl (–SMe) anchor at one end and a freshly generated pyridine anchor at the other. We then compared the conductance of the **PyBz-*n*** compound with that of an ex-situ synthesized **Py-*n*** compound under high bias voltage. This comparison provides crucial evidence supporting the SET-induced homolytic cleavage of the C–N bond in Katritzky salts (**PyBz-*n***).

- 7. *The beta quoted in SI fig 17 is not correct. The authors seem to use plateau length since they claim that PyBz-1 has a length of 0.2 nm while PyBz-3 has a length of 1.3 nm. Also, they seem to be fitting the beta plot on a base-10 log scale, not a natural log scale which is again completely wrong.*

Reply

Thank you for your reminder. We were really sorry for our careless mistakes. We have revised the value accordingly. Please see changes in **Supplementary Fig. 18** on Page S23 in SI.

- 8. The authors showed the flicker noise power results in Fig. 2e. We believe that the author should also provide the raw power spectral density before FFT. Also, did the authors select conductance traces for their flicker-noise data analysis? What are the selection criteria? How many traces are selected? This information should also be included in the manuscript.

Reply

According to the reviewer's comments, we will upload the raw data to make our results convincing. In addition to removing curves that did not hover successfully and undergo direct tunneling, no other selecting method was applied.

- 9. In Supp. Fig. 24, the authors showed their CV data for PhTMP-n using glassy carbon as working electrode, Ag/AgCl as reference electrode, and Pt as counter electrode. For $n = 1-3$, the measured CVs exhibit irreversible reduction process at different potentials. However, these results do not necessarily reflect what happens in molecular junctions with gold electrodes, as gold has a different reactivity from glassy carbon. The authors should instead measure in-situ CVs using their STM-BJ setup.

Reply

We thank this reviewer for bringing up the concern of “*in-situ CVs*”. Performing *in situ* cyclic voltammetry (CV) experiments in our STM-BJ set-up is challenging due to the inability to completely eliminate oxygen from the junction. As a result, we opted to use an Au electrode instead of a glassy carbon electrode as the working electrode for the CV experiments in the electrochemical workstation.

As shown in **Figure R15**, when the Au disk electrode is used as the working electrode, the reduction potential of **PhTMP-1** is approximately -0.9 V, which is shifted by 0.2 V compared to the potential obtained by glassy carbon electrode. We believe that the adsorption of the compounds on Au electrode may facilitate electron transfer, thus lowering the reduction potential.

Figure R15. Cyclic voltammograms (CV) experiments performed on PhTMP-1.

The CV experiments were performed in MeCN with potassium tetrafluoroborate (KBF_4 , 1 mM) as the auxiliary electrolyte. Working electrode (WE): Au disk; Reference electrode (RE): Ag/AgCl; Counter electrode (CE): platinum wire.

Listed below are the details of our responses to the referees' comments.

Reviewer #1(Remarks to the Author):

- *1. A pleasure to read and a significantly improved manuscript.*

Reply

Thanks very much for this esteemed Reviewer's positive feedback and for acknowledging the improvements in our revised manuscript. We are grateful for the recognition of our efforts in enhancing the manuscript. The comments have been invaluable in helping us refine our work.

- *2. Only one question: why potassium tetrafluoroborate as supporting salt? It is very unusual to see this used as electrolyte in MeCN since it very limited solubility. This forced the authors to add a salt at 1 mM concentration where the Debye length will be significantly larger than the molecule hence poor fields near the electrode, leading to slow kinetics of electron transfer.*

Reply

We appreciate the Reviewers' attention to the solubility of KBF₄ in acetonitrile (MeCN) and its impact on the ionic strength of the solution.

Firstly, the primary reason for selecting KBF₄ is its compatibility with the counterion we use during the synthesis in **PhTMP-*n*** (BF₄⁻) under study. Although its solubility is limited, KBF₄ helps to avoid anion exchange during the cyclic voltammograms (CV) experiments.

As suggested by this Reviewer, we have replaced KBF₄ with Tetrabutylammonium hexafluorophosphate (TBAPF₆) as the supporting electrolyte during the CV experiments of **PhTMP-*n***. Using **PhTMP-1** as an example, which was shown in **Figure R1**, we found that electrolyte substitution has a minimal impact on the reduction potential but a significant effect on the current magnitude.

In summary, by supplementing with TBAPF₆, a supporting electrolyte with better solubility, we aim to address the reviewers' concerns regarding the choice of potassium tetrafluoroborate (KBF₄) as the supporting electrolyte and its limited solubility.

Figure R1. Cyclic voltammograms (CV) experiments performed on PhTMP-1. The CV experiments were conducted in MeCN using a Au working electrode (WE), an Ag/AgCl reference electrode (RE), and a platinum wire counter electrode (CE). The black curves represent experiments with potassium tetrafluoroborate (**KBF₄**, 10 mM) as the auxiliary electrolyte, while the red curves correspond to experiments with tetrabutylammonium hexafluorophosphate (**TBAPF₆**, 100 mM) as the supporting electrolyte.

Reviewer #2 (Remarks to the Author):

- 1. *The authors have carefully addressed my comments providing additional information that accurately presents the current state of the art in the field and additional experimental and theoretical proofs that support their findings. I strongly encourage the inclusion of figures R5, R6, R7 and the related explanations at least into the Supplementary Information. This material is essential and supports the interpretation of the results.*

Reply

Thanks very much for this reviewer's thorough review and for the valuable suggestions. We fully agree with the recommendation regarding the inclusion of figures R5, R6, and R7, along with the related explanations to the supporting information. As suggested, we added these figures and their detailed explanations into **Supplementary Fig. 36-38** on **Page S38-40** in the SI. We believe that this additional material will provide important context and further support the interpretation of our results. Thanks again for this Reviewer's constructive feedback and encouragement. We hope that the inclusion of this supplementary material will further strengthen the manuscript.

- 2. *However, the inclusion of the Au/PhTMP-n/GaOx/EGaIn system in the main text, is still not convincing. These junctions do not have any relation to the main message of the manuscript that addresses the electrocatalytic Au-C bond formation. Moreover, the envisaged rectification effect of the asymmetric coupling (Au-S vs. Au-C) of a molecular system should be demonstrated in junctions with similar electrodes not in asymmetric junctions with different electrodes. Besides, the experimental system Au/PhTMP-n/GaOx/EGaIn is different from the Au/Ph-n/Au considered for theoretical calculations.*

Reply

We thank this Reviewer for the thoughtful comment regarding removing the Au/PhTMP-n/GaOx/EGaIn part in the main text. We appreciate your observation that the rectification effect of asymmetric coupling (Au-S vs. Au-C) should be demonstrated in junctions with similar electrodes. We have therefore removed this

section to align experimental focus with our computational framework (Au/Ph-n/Au systems). The revision sharpens mechanistic connections between electrocatalytic Au–C bond formation and transport signatures, eliminating confounding variables from heterogeneous interfaces. We will ensure that the revised manuscript presents a clearer connection between the experimental and theoretical results, strengthening the focus on Au–C bond formation in electrocatalysis. Thank you again for your insightful comments. We believe these revisions will help enhance the clarity and coherence of the manuscript.

➤ *3. After addressing the final comments, I recommend the acceptance of the manuscript for publication.*

Reply

Thank you for your valuable feedback and for recommending our manuscript for acceptance after addressing the final comments. We deeply appreciate your time and effort in reviewing our work.

Reviewer #3 (Remarks to the Author):

➤ *1. Authors reviewed the manuscript adequately. I recommend publication.*

Reply

We thank the Reviewer for the favorable assessment and publication recommendation, which acknowledges the enhanced methodological rigor and conceptual precision achieved through our revision process.

Reviewer #4 (Remarks to the Author):

- 1. *The authors have attempted to address a few of the concerns raised however, from their revision, it is even more clear that they have not achieved their main claim, i.e. to create Au-C linked molecular junction starting with the pyridine salts. More importantly, they have simply not addressed many points in the review. Additionally, some new data added further weakens the novelty of this work.*

Reply

We carefully examined the Reviewers' comments and spent over six months adding a large number of additional controlled experiments to provide more convincing evidence. We hope that our revisions will make the manuscript more rigorous and comprehensive.

- 2. *As stated in the introduction of their manuscript, “The stability of Au–C covalent bonds in air and water, along with their unique electronic properties, make them highly advantageous for achieving [6-9] stronger coupling strength compared to other commonly used anchor groups such as amines¹⁰, ¹¹, thiols [12], thiomethyls[13], pyridines [14, 15], etc.” In their revision, they argue that the reason they observe a low conductance for the PhTMP series is because they have a methylene group which they claim reduces the conductance. The entire point of Au-C links is to have a high conductance. If they cannot achieve this, then this manuscript has limited novelty.*

Reply

We are truly grateful for your meticulous review and the insightful comments you've provided. While we understand your concerns, we'd like to offer some clarifications that we believe add important context to our work.

To establish mechanistic controls, we synthesized **PhSn–n** series featuring methylthio anchoring groups on one terminus and tributyltin moieties on the opposing end (**Figure R2**). We sincerely appreciate the Reviewer's insightful comment regarding the orbital matching between Au and sp³-type methylene carbon. This observation has prompted us to re-examine our system, leading to the recognition that the observed low conductivity in the Au–S–Ph³–C–Au molecular series likely stems from an asymmetric orbital alignment where only the C–Au terminal achieves effective orbital matching

with Au electrodes. Such unilateral alignment creates an electronic transport bottleneck, significantly hindering efficient electron propagation through the molecular backbone.

This crucial understanding has been incorporated into the revised discussion accompanying **Figure 3** in the main text on Page 11 to better elucidate the structure-property relationship in these molecular junctions.

Figure R2. **a**, PhTMP- n series. **b**, PhSn- n series. **c**, The 1D conductance histograms of PhTMP- n recorded at a bias voltage of 0.6 V (top) and PhSn- n datasets recorded a bias voltage of 0.1 V (bottom). **d**, The 2D conductance histograms of PhTMP-3 recorded at bias voltages of 0.6 V (top), and the 2D conductance histograms of PhSn-3 recorded at bias voltages of 0.1 V (bottom). **e**, Calculated transmission spectra for PhTMP-1 under increasing bias voltages from 0 to 0.6 V. Inset shows the position of transmission eigenstate corresponding to Au-C gateway state at 0.32 eV.

➤ 3. They claim that the low conductance observed in their junctions is Au-C terminated junctions by comparing the conductance of a new molecule SCh2-1 which has a lower conductance compared to S-1. However, this completely misses the point of using C-as the linker. When you create an Au-S-CH2-Ph linked system, there are two atoms between the phenyl and Au (S and CH2). However, for the Au-CH2-Ph, there is only one linker atom, the carbon. Their argument is therefore completely not valid. Secondly, In Chen et al, 133, 17160–17163 (2011), the conductance of the entire series is very high because there is a Au-CH2-Ph linker (their table in the SI – scheme 1 - is incorrect.) Without the CH2 group, the Au does not couple into the pi-system and conductances are lower. The bottom line is that

these authors have not demonstrated the creation of Au-C linked molecular junctions.

Reply

We apologize for the oversight in our analysis regarding the citation of Chen et al. (*J. Am. Chem. Soc.* **2011**, *133*, 17160–17163). We have made the necessary corrections in the revised version of manuscript.

As suggested by the Reviewer and Editor, we have spent another six months to synthesize the **Ph(TMP)_{2-n}** series of molecules, which can form Au-C bonds on both sides of the molecule (**Figure R3**). Synthetic and analytical details of the TMP-terminated **Ph(TMP)_{2-n}** series are detailed in **Supplementary Sections B and C**.

However, the large steric hindrance of TMP groups on both ends of **Ph(TMP)_{2-n}** compromises electrode-molecule coupling during the tip-substrate opening process in STM-BJ experiments, significantly reducing SET efficiency and preventing effective *in situ* formation of Au–C–**Phⁿ**–C–Au junctions. In contrast to the junction opening process, tip-mediated mechanical force applied during the junction closing process drives the contacts between the tips and TMP groups, thereby enhancing SET efficiency while facilitating *in situ* Au–C bonds formation at the interface.

Fig. R3b–d display the 2D conductance-displacement histograms of the closing process for Au–C–**Phⁿ**–C–Au junctions formed *in situ* from **Ph(TMP)_{2-n}** ($n = 1, 2,$ and 3) precursors. The relative displacement distributions for Au–C–**Phⁿ**–C–Au series obtained from the closing process are demonstrated in **Fig. R3b–d** insets. Considering a jump-to-contact behavior of Au electrodes during the closing process with a 0.2 nm calibration distance applied, the molecular lengths derived from Au–C–**Phⁿ**–C–Au ($n = 1, 2,$ and 3) are 0.8, 1.2, and 1.6 nm, respectively. From the 1D conductance histogram (**Fig. 4e**), it is evident that Au–C–**Phⁿ**–C–Au junctions with Au–C covalent bonds at both ends show *high-conductance values* reaching 10^0 , $10^{-1.0}$, and $10^{-1.7} G_0$ — with one, two, and three phenylene rings in the backbones, respectively.

Figure R3. Formation of Au–C covalent bonds at both termini in single-molecule junctions. **a**, Schematic illustration showing the electrocatalytic formation of benzyl-type Au–C bonds at both termini of during the closing processes. **b–d**, The 2D conductance-displacement histograms constructed from the closing processes for the Au–C–Phⁿ–C–Au junctions. Insets show the corresponding relative distance distributions. **e**, 1D conductance histograms constructed from the closing processes showing the dominant conductance signals for the Au–C–Phⁿ–C–Au junctions. **f**, Calculated transmission spectra for Au–C–Phⁿ–C–Au junctions. **g**, Conductance versus molecular length determined from experimental measurements (red dots), theoretical calculations (red circle), and benchmark from reference by Chen et al. (*J. Am. Chem. Soc.* **2011**, *133*, 17160–17163).

- 4. The authors have not addressed why the beta value obtained for the PhTMP-*n* series is very low. This is a critical issue. To prove that you can measure a series with a new linker, the first step is to show that the series gives a correct conductance decay. This has not been done. They also do not provide a clean comparison of conductance vs length for this series.

Reply

Thank you for your careful review of our work and for providing detailed feedback on the conductivity attenuation coefficients of the **PhTMP-*n*** series. To address the issues raised, we synthesized a series of **PhSn-*n*** molecules with an identical molecular skeleton, featuring terminal tin groups (**Figure R2**). The molecular conductance and attenuation coefficients of the **PhTMP-*n*** and **PhSn-*n*** series are very similar (**Figure R2**), which confirms that our attenuation coefficients fall within a reasonable range. We hope this response addresses the Reviewer's concerns.

The observed low conductance in the **PhTMP-*n*** series warrants further investigation, and we plan to systematically examine its origin in subsequent studies. We thank the Reviewer for this constructive feedback and will address these points comprehensively in our revised submission.

Figure R4. Experimental conductance values (**PhTMP-*n***: red solid; **PhSn-*n***: black open) derive from 1D Gaussian fits of the corresponding molecular junctions.

- 5. *The new x-ray data added shows carbon peaks even on clean gold – this is a major problem. They do not have clean Au surfaces and thus the rest of their data cannot simply be attributed to the system they study.*

Reply

We fully understand the concern that the presence of carbon peaks on clean gold surfaces could indicate contamination, which would complicate the interpretation of our data. XPS is not a reliable technique for characterizing the in situ formation of Au–C bonds. In the revised version, we have completely removed all discussions and conclusions related to the self-assembled monolayers.

- 6. *The authors have not shown any FFT data in the manuscript or SI.*

Reply

The FET source data (Figure R5) have been deposited in the *Nature Communications* repository (<https://figshare.com/s/4f3e8f650223434adc7b>) and are freely accessible for reviewer verification.

- 7. *If the authors indeed have a new way to create Au-C bonded junctions, they should demonstrate this with both sides of the molecule linked with Au-C bonds. With their method, this is not possible however with halide terminations, this is trivial. The authors claim that the problem with halide terminations is that it can lead to dimerization. This is correct but in their system, they should also see dimerization. Any method that creates a benzyl radical as an intermediate has the possibility to dimerize. This is the case with halides, SnMe3 and any other pathway. The fact that they now claim they do not see dimerization indicates that they do not create the radical species as an intermediate.*

Reply

Firstly, As suggested by the Reviewer and Editor, we have spent another six months to synthesize the **Ph(TMP)_{2-n}** series of molecules, which can form Au-C bonds on both sides of the molecule (**Figure R5**). Synthetic and analytical details of the TMP-terminated **Ph(TMP)_{2-n}** series are detailed in **Supplementary Sections B** and **C**.

However, the large steric hindrance of TMP groups on both ends of **Ph(TMP)_{2-n}** compromises

electrode-molecule coupling during the tip-substrate opening process in STM-BJ experiments, significantly reducing SET efficiency and preventing effective *in situ* formation of Au–C–Phⁿ–C–Au junctions. In contrast to the junction opening process, tip-mediated mechanical force applied during the junction closing process drives the contacts between the tips and TMP groups, thereby enhancing SET efficiency while facilitating *in situ* Au–C bonds formation at the interface.

Fig. R5. Formation of Au–C bond through SET-induced homolysis of Ph(TMP)₂-3 in a single-molecule junction.

Secondly, we systematically compared two distinct synthetic approaches for constructing Au–C(sp²) bonds: one employing tin-containing terminal groups versus another utilizing Katritzky salts. It's arranged below:

	Katritzky salts	tin compound
Toxicity	hypotoxicity	High toxicity
Product purity and purification	Impurities are less and can be purified by ion exchange	Intractable purification & trace impurity-mediated Au-Au disappear
stability	High stability, not easy to occur side reactions	Poor stability, the side reactions are easy to occur in the post-treatment
Controllability of reaction	Threshold voltage is required to activate the reaction & highly controllable	Unstable itself & lose to control accurately

I hope these answers address your concerns.

Reviewer #5 (Remarks to the Author):

- *I co-reviewed this manuscript with one of the reviewers who provided the listed reports. This is part of the Nature Communications initiative to facilitate training in peer review and to provide appropriate recognition for Early Career Researchers who co-review manuscripts.*

Listed below are the details of our responses to the referees' comments.

Reviewer #2(Remarks to the Author):

- *1. The authors have carefully revised the manuscript, removing the unrelated information on self-assembled monolayers and including additional experimental results related to the formation of covalent Au–C contacts at both molecular termini. These results demonstrate a significant enhancement in conductance.*

Reply

We are grateful for the recognition of our efforts in enhancing the manuscript. The comments have been invaluable in helping us refine our work. Thanks very much for the Reviewer's positive feedback again.

- *2. Regarding the new data, I recommend specifying the applied bias voltage for the different measurements presented in Figure 4.*

Reply

Thank you for this helpful suggestion. We have now specified the applied bias voltages used for each measurement in Figure 4. This information has been added to the main text on page 14, to ensure better clarity and reproducibility.

- *3.As a minor suggestion to support visual interpretation and facilitate differentiation between the presented systems, I propose reversing the displacement axis in Figure 4b (i.e., displaying decreasing displacement values). This would reflect the actual measurement procedure, which involved approaching the tip toward the surface (“tip–surface closing process”), in contrast to the other systems discussed in the manuscript.*

Reply

Following your recommendation, we have adjusted the position of the axes in the figure. However, we found that the revised version appears less clear and may potentially cause confusion for readers. We are concerned that this modification may impair data clarity for the molecular electronics community. Therefore, after careful consideration, we have decided to retain the original axis placement to preserve the clarity of the figure (another work using the closing process of STM-BJ: Liu, J. *et al. Chem* **2019**, *5*, 390–401). We hope this decision is acceptable and are happy to make further adjustments if needed.

Figure R1 | Formation of Au–C bond through SET-induced homolysis of Ph(TMP)₂-*n* (*n* = 1, 2, and 3) in a single-molecule junction.

➤ *4. Once the final comments are addressed, I believe the manuscript is suitable for publication.*

Reply

We sincerely appreciate your positive assessment and are grateful for your constructive feedback throughout the review process. We have carefully addressed the remaining comments, and we hope the revised manuscript meets your expectations.

Reviewer #4 (Remarks to the Author):

➤ *1. I appreciate the authors for taking my comments seriously and dedicating time to address them. Their PhSn-*n* control experiments convincingly demonstrate the formation of molecular junctions with Au-C covalent bonds. Furthermore, they showed that they were able to form Au-C bonds with both electrodes using Ph(TMP)₂-*n*, leading to an exponential decay in great agreement with previous findings, which reinforces the significance and novelty of this work.*

Reply

We sincerely thank the reviewer for the encouraging and thoughtful feedback. We truly appreciate your recognition of the significance and novelty of our work, and we are grateful for your constructive input throughout the review process.

➤ *2. According to the manuscript, SET is the rate-limiting step of Au-C formation, which implies the steady-state concentration of benzyl radical is very low, thereby reducing the likelihood of radical dimerization. However, it would still be nice to include another control experiment using a radical scavenger (such as butylated hydroxytoluene). If the presence*

of a scavenger inhibits or blocks the Au-C junction formation, it would be helpful on diagnosing the mechanism.

Reply

We thank the reviewer for this insightful suggestion. In response, we have performed the additional control experiment using butylated hydroxytoluene (BHT) as a radical scavenger. We added 10 equiv. of BHT to our standard reaction mixture under the same reaction conditions used in the main text. These results are shown in **Figure R2**.

Upon the addition of BHT, three possible products may be formed due to the unclear reaction mechanism. The initial product likely features weak Au- π or π - π interactions (similar work see: *Nano Lett.* **2023**, *23*, 567). We can only observe faint and broad signals in both the 1D and 2D conductance-displacement histograms.

Furthermore, substantial steric hindrance in the latter two products likely impedes Au-molecule contact at the interface, preventing stable anchoring. Although product identity remains unresolved, BHT scavenging inhibits Au-C bond formation under these conditions, suppressing characteristic conductance signatures.

Figure R2 | Effect of BHT radical scavenger on Au-C bond formation. Comparison of product yields under standard reaction conditions with and without 10 equiv. BHT.

- 3. The authors show some noise analysis (Figure 2e) however there are no details on how this noise is calculated. Either the authors should show raw data that demonstrates how they make these images or simply remove this panel from the figure as it does not really add any new information.

Reply

We appreciate the reviewer's insightful comment. To avoid any confusion that this panel does not significantly enhance the overall findings, we have decided to remove **Figure 2e** from the revised manuscript.

- 4. The claim in the abstract that the conductance of the 8Å wire is $1 G_0$ is a bit misleading. Figure 4e shows a dark black fit over the data which hides the raw data behind it. From the 2D histogram in Figure 4b, it is clear that the peak is below 0. They should not overstate the findings with the statement "a record value for molecular wires of comparable length" since this value is basically the same as found by earlier researchers.

Reply

We thank the reviewer for this important observation. We agree that the presentation of the data in Figure 4e could be improved for clarity. We have revised the abstract to reflect this more accurately, now stating that the 8 Å wire shows "a conductance approaching $1 G_0$ " and we have removed the phrase "a record value for molecular wires of comparable length" to avoid overstating the novelty. Instead, we now emphasize that the result is "comparable to the highest reported values for molecular wires of similar length" on **Page 2** in main text, aligning our discussion with existing literature.

We appreciate the reviewer's careful attention to this issue and believe these changes lead to a more precise and balanced presentation of our findings.

- 5. The 2D histograms in Figure 4 all show some vertical lines. Is this a resolution problem?

Reply

Thank you for your observation. We have adjusted the resolution of Figure 4 to improve its clarity. The updated version provides a clearer visualization, and we believe it addresses the issue you raised. The revised figure has been included in the manuscript (see revised Figure 4 on page 14).

Listed below are the details of our responses to the referees' comments.

Reviewer #4 (Remarks to the Author):

➤ *1. The authors have addressed all concerns raised.*

Reply

Thank you very much for the reviewer's confirmation that all concerns have been addressed. We greatly appreciate the valuable time and efforts you have dedicated to reviewing our manuscript, as well as the constructive feedback that has significantly helped improve the quality of our work.